# LEARNING REPRESENTATIONS OF INSTRUMENTS FOR PARTIAL IDENTIFICATION OF TREATMENT EFFECTS

## ABSTRACT

Reliable estimation of treatment effects from observational data is important in many disciplines such as medicine. However, estimation is challenging when unconfoundedness as a standard assumption in the causal inference literature is violated. In this work, we leverage arbitrary (potentially high-dimensional) instruments to estimate bounds on the conditional average treatment effect (CATE). Our contributions are three-fold: (1) We propose a novel approach for partial identification through a mapping of instruments to a discrete representation space so that we yield valid bounds on the CATE. This is crucial for reliable decision-making in real-world applications. (2) We derive a two-step procedure that learns tight bounds using a tailored neural partitioning of the latent instrument space. As a result, we avoid instability issues due to numerical approximations or adversarial training. Furthermore, our procedure aims to reduce the estimation variance in finite-sample settings to yield more reliable estimates. (3) We show theoretically that our procedure obtains valid bounds while reducing estimation variance. We further perform extensive experiments to demonstrate the effectiveness across various settings. Overall, our procedure offers a novel path for practitioners to make use of potentially high-dimensional instruments (e.g., as in Mendelian randomization).

## 1 INTRODUCTION

Estimating the conditional average treatment effect (CATE) from observational is an important task for personalized decision-making in medicine (Feuerriegel et al., 2024). For example, a common question in medicine is to estimate the effect of alcohol consumption on the onset of cardiovascular diseases (Holmes et al., 2014). There are many reasons, including costs and ethical concerns, why CATE estimation is often based on observational data (such as, e.g., electronic health records, clinical registries).

However, identifying the CATE from observational data is challenging as it typically requires *strong* assumptions in the form of *unconfoundedness* (Rubin, 1974). Unconfoundedness assumes there exist no additional unobserved confounders $U$ between treatment $A$ and outcome $Y$. If the unconfoundedness assumption is violated, a common strategy is to leverage **instrumental variables (IVs)** $Z$. IVs affect only the treatment $A$ but exclude unobserved confounding between $Z$ and $Y$, which often can be ensured by design such as for randomized studies with non-compliance (Imbens & Angrist, 1994). The causal graph for the IV setting is shown in Fig. 1.

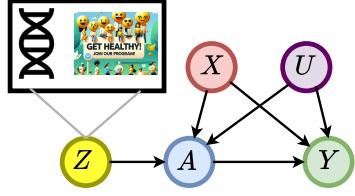

Figure 1: Overview of the IV setting. We consider complex instruments $Z$ (e.g., gene data, text, images), observed confounders $X$, unobserved confounders $U$, a binary treatment $A$, and an outcome $Y$.

**Motivational example:** *Mendelian randomization.* Mendelian randomization (Pierce et al., 2018) refers to the use of genetic information as instruments $Z$ to estimate the effect of a treatment or exposure $A$ (e.g., alcohol consumption) on some medical outcome $Y$ (e.g., cardiovascular diseases). In this setting, there are further patient characteristics that are observed ($X$) but also unobserved ($U$), which one accounts for through the instrument. Yet, common challenges are that **(i)**

instruments with genetic information are often *high-dimensional* and **(ii)** involve *complex, non-linear relationships between instruments and treatment intake or exposure*.

However, existing IV methods using machine learning for point estimation of the CATE rely on *strong simplifying assumptions* ($\rightarrow$ violating **(ii)** from above). For example, some methods assume linearity in some feature space in the CATE and make other, strict parametric assumptions on the unobserved confounders such as additivity or homogeneity (Hartford et al., 2017; Singh et al., 2019; Xu et al., 2021). Yet, such simplifying assumptions are often *not* realistic and can even lead to unreliable and false conclusions by the mis-specification of the CATE.

A potential remedy is to use IVs for **partial identification** of the CATE where one circumvents any hard parametric assumptions by estimating upper and lower bounds of the CATE (Manski, 1990). This is usually sufficient in medical practice when one is merely interested in whether a treatment variable (e.g., exposure as in Mendelian randomization) has a positive or a negative effect. So far, methods for partial identification of the CATE in IV settings are rare. There exist closed-form bounds (i.e., via a fixed target estimand that can be learned), yet only for the setting with both *discrete* instruments and *discrete* treatments (Balke & Pearl, 1997).

Existing machine learning methods for partial identification are typically designed for *simple* instruments that are binary or discrete ($\rightarrow$ violating **(i)** from above). Alternatively, methods that extend partial identification for continuous instruments require *unstable* training paradigms such as adversarial learning (Kilbertus et al., 2020; Padh et al., 2023) which becomes even more unstable for more complex instruments. In contrast, there is a scarcity of methods that can deal robustly with continuous, as well as *complex* and potentially high-dimensional instruments such as, e.g., gene expressions as in Mendelian randomization but also text, images, or graphs.[1]

**Our paper:** In this work, we leverage complex instruments for partial identification of the CATE. Specifically, we allow for instruments that can be continuous and potentially high-dimensional (such as gene information) and, on top of that, we explicitly allow for complex, non-linear relationships between instruments and treatment intake or exposure. In the rest of this paper, we refer to this setting as "complex" instruments.

To this end, we proceed as follows. (1) We propose a novel approach for partial identification through a mapping of complex instruments to a discrete representation space so that we yield valid bounds on the CATE. We motivate our approach in Fig. 2. (2) We derive a two-step procedure that learns tight bounds using a neural partitioning of the latent instrument space. As a result, we avoid instability issues due to numerical approximations or adversarial training, which is a key limitation of prior works. We further improve the performance of our procedure by explicitly reducing the estimation variance in finite-sample settings to yield more reliable estimates. (3) We provide a theoretical analysis of our procedure and perform extensive experiments to demonstrate the effectiveness across various settings.

**Contributions:**[2] (1) To the best of our knowledge, this is the first IV method for partial identification of the CATE based on complex instruments. (2) We derive a two-step procedure to learn tight bounds. (3) We demonstrate the effectiveness of our method both theoretically and numerically.

## 2 RELATED WORK

**Machine learning for CATE estimation with IV:** Existing works have different objectives. One literature stream leverages IVs for CATE estimation but focuses on settings where the treatment effect can be identified from the data. This includes work that extends the classical two-stage least-squares estimation to non-linear settings by learning non-linear feature spaces (Singh et al., 2019; Xu et al., 2021), deep conditional density estimation in the first stage (Hartford et al., 2017), or using moment conditions (Bennett et al., 2019). Another literature stream aims at new machine learning methods with favorable properties such as being doubly robust (Kennedy et al., 2019; Ogburn et al., 2015; Semenova & Chernozhukov, 2021; Syrgkanis et al., 2019) or multiply robust (Frauen & Feuerriegel, 2023).

---

[1]In Appendix B, we provide an extended discussion about the real-world relevance of our method.
[2]Both codes and data are available via `https://anonymous.4open.science/r/ComplexPartialIdentification-2500/`.

Recently, researchers started applying machine learning methods to IVs from Mendelian randomization (Legault et al., 2024; Malina et al., 2022), which is our motivational example from above. However, these works aim at point-identified CATE estimation with IVs. As a result, these rely on *hard* and generally untestable assumptions on some effects in the causal graph, such as linearity, monotonicity, additivity, or homogeneity (Wang & Tchetgen Tchetgen, 2018). This is unlike our method for partial identification that does *not* require such hard assumptions and that is non-parametric.

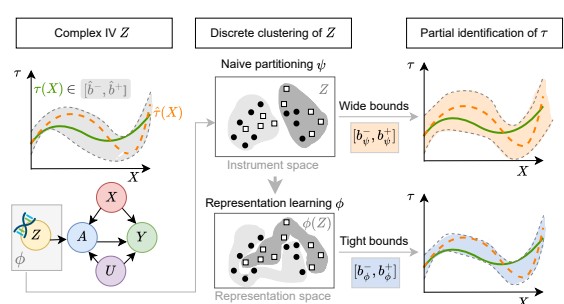

Figure 2: Leveraging complex instruments for partial identification of the CATE through discrete representations of $Z$. Naïve discretization on the IV input space leads to wide, and thus non-informative, bounds. Our method learns a latent representation $\phi(Z)$ to yield tight bounds.

**Partial identification:** Partial identification aims to identify and learn upper and lower bounds of some causal quantity (e.g., the CATE) when the causal quantity itself cannot be point identified from the data and assumptions. In a general setting with binary treatments, Robins (1989) and Manski (1990) derived closed-form bounds on the ATE for bounded outcomes $Y$. Further work extended these ideas to settings with binary instrumental variables, binary treatments, and binary outcomes (Balke & Pearl, 1994; 1997) to derive tighter bounds. Newer approaches for discrete variables include the works of Duarte et al. (2023) and Guo et al. (2022). Swanson et al. (2018) provide an extensive overview of partial identification in this setting. Other works focus on how to leverage additional observed confounders to further tighten bounds on the ATE (see, e.g., Levis et al., 2023). However, these works do not focus on efficiently leveraging continuous or even high-dimensional instruments for learning tight bounds, unlike our work that is tailored to such complex instruments.

Another literature stream focuses on partial identification under general causal graphs (Balazadeh et al., 2022), including IV settings with continuous variables such as continuous treatments (Gunsilius, 2020; Hu et al., 2021; Kilbertus et al., 2020; Padh et al., 2023). However, these methods either make strong assumptions about the treatment response functions or require unstable optimization via adversarial training and/or generative modeling such as through using GANs. This can easily result in *unreliable* estimates of bounds for finite data, especially with high-dimensional instruments. Further, these methods are *not* directly tailored for binary treatments, unlike our method.

**Research gap:** To the best of our knowledge, reliable machine learning methods for partial identification of the CATE with complex instruments are missing. To draw conclusions about CATEs (as in, e.g., Mendelian randomization), our method is the first to: (i) make use of the complex instrument information (e.g., continuous or high-dimensional), (ii) avoid making strong parametric assumptions by focusing on partial identification, and (iii) avoid unstable training procedures such as adversarial learning.

## 3 PROBLEM SETUP

**Setting:** We focus on the standard IV setting (Angrist et al., 1996; Wooldridge, 2013). Hence, we consider instruments (e.g., gene data, text, images) given by $Z \in \mathcal{Z} \subseteq \mathbb{R}^d$ but, unlike previous research, allow the instruments to be complex. As such, we allow the instruments to be continuous and potentially high-dimensional. We further have access to an observational dataset $\mathcal{D} = \{z_i, x_i, a_i, y_i\}_{i=1}^n$ of size $n$. The data is sampled i.i.d. from a population $(Z, X, A, Y) \sim \mathbb{P}$, with observed confounders $X \in \mathcal{X} \subseteq \mathbb{R}^p$, binary treatments $A \in \mathcal{A} \subseteq \{0,1\}$, and bounded outcomes $Y \in \mathcal{Y} \subseteq [s_1, s_2] \subseteq \mathbb{R}$. Additionally, we allow for unobserved confounders $U$ of arbitrary form between $A$ and $Y$.

We further assume a causal structure as shown in Fig. 1. In particular, we assume that $Z$ is an instrumental variable that has an effect on the treatment $A$ but no direct effect on the outcome $Y$ except through $A$. Further, we assume that $Z$ is independent of $X$, e.g., by randomization. In Appendix B, we provide an extended discussion to show the real-world relevance and validity of our assumptions in different settings.

**Notation:** Throughout our work, we denote the *response function* by $\mu^a(x,z) := \mathbb{E}[Y|X = x, A = a, Z = z]$ and the *propensity score* by $\pi(x,z) := \mathbb{P}(A = 1|X = x, Z = z)$.

**CATE:** We use the potential outcomes framework (Rubin, 1974) to formalize our causal inference problem. Let $Y(a) \in \mathcal{Y}$ denote the potential outcome under treatment $A = a$. We are thus interested in the CATE $\tau(x) = \mathbb{E}[Y(1) - Y(0)|X = x]$.

**Identifiability:** We make the following standard assumptions from the literature in partial identification with IVs (Angrist et al., 1996). **Assumption 1** (*Consistency*): $Y(A) = Y$. **Assumption 2** (*Exclusion*): $Z \perp\!\!\!\perp Y(A) \mid (X, A, U)$. **Assumption 3** (*Independence*): $Z \perp\!\!\!\perp (U, X)$.

Note that, however, Assumptions 1–3 from the standard IV setting are *not* sufficient to ensure identifiability of the CATE (Gunsilius, 2020). To ensure identifiability, one would require additional assumptions, such as linearity or, more generally, additive noise assumptions (Hartford et al., 2017; Wang & Tchetgen Tchetgen, 2018). Yet, such assumptions are highly restrictive and are neither testable nor typically ensured in real-world scenarios. Hence, this motivates our objective to perform partial identification instead.

**Objective**: We frame our objective as a *partial identification* problem and thus focus on estimating *valid* bounds $(b^-(x), b^+(x))$ for the CATE $\tau(x)$ such that $b^-(x) \leq \tau(x) \leq b^+(x)$ holds for all possible $x \in \mathcal{X}$. Furthermore, the bounds should be *informative*, i.e., we would like to minimize the expected bound width $\mathbb{E}_X[b^+(X) - b^-(X)]$, while still ensuring validity. Formally, we aim to solve

$$b_*^-, b_*^+ \in \underset{b^-, b^+}{\arg\min} \, \mathbb{E}_X[b^+(X) - b^-(X)] \quad \text{s.t.} \quad b^-(x) \leq \tau(x) \leq b^+(x) \quad \text{for all} \quad x \in \mathcal{X}. \quad (1)$$

## 4 PARTIAL IDENTIFICATION OF CATE WITH COMPLEX INSTRUMENTS

### 4.1 OVERVIEW

We now present our proposed method to solve the partial identification problem from Eq. (1). Solving Eq. (1) directly is *infeasible* because it involves the unknown CATE $\tau(x)$. Hence, we propose the following approach:

> **Outline:** ① We learn a discretized representation (also called partitioning) $\phi(Z)$ of the instrumental variable $Z$. ② We then derive closed-form bounds given the discrete representation $\phi$. ③ We transform the closed-form bounds back to our original bounding problem and, in particular, express all quantities involved as quantities that can be estimated from observational data.

Below, we first explain why existing closed-form bounds are *not* directly applicable and why deriving such bounds is non-trivial. We then proceed by providing the corresponding theory for the above method. Specifically, we first take a population view to show theoretically that our bounds are valid (Sec. 4.2). Then, we take a finite-sample view and present an estimator (Sec. 4.3).

*Limitations of existing bounds:* There exist different approaches for bounding treatment effects (see Sec. 2) using continuous instruments, yet these either require additional assumptions or can easily become unstable, especially for high-dimensional $Z$. Furthermore, these bounds consider continuous treatments but are *not tailored* for binary treatments (e.g., whether a drug is administered). Hence, we derive custom bounds for our setting.

*Why is the derivation non-trivial?* For binary treatments, it turns out that there exist closed-form solutions for bounds whenever the instrument $Z$ is discrete. That is, the existing bounds for the average treatment effect (ATE) with continuous bounded outcome proposed in (Manski, 1990) can be extended to non-parametric closed-form bounds for the CATE (Schweisthal et al., 2024). While these bounds are useful in a setting with discrete instruments $Z$, they are not directly applicable to continuous or even high-dimensional $Z$ due to two main reasons: (1) The bounds need to be evaluated for *all* combinations $l, m \in \mathcal{Z}^2 \subseteq \mathbb{R}^d \times \mathbb{R}^d$, which is *intractable*. (2) Evaluating the bounds only on a random subset of combinations $l, m$ can result in *arbitrary high* estimation variance for regions with a low joint density of $p(X = x, Z = l)$ or $p(X = x, Z = m)$. Hence, we must derive a novel method for estimating bounds based on complex instruments (that are, e.g., continuous or high-dimensional), yet this is a highly *non-trivial* task.

## 4.2 POPULATION VIEW

In the following theorem, we provide a novel theoretical result of how to obtain valid bounds based on discrete representations $\phi(Z)$ of the instrument $Z$.

**Theorem 1** (Bounds for arbitrary instrument discretizations)**.** *Let $\phi : \mathcal{Z} \to \{0, 1, \ldots, k\}$ be an arbitrary mapping from the high-dimensional instrument $Z$ to a discrete representation. We define*

$$\mu_\phi^a(x, \ell) = \int_Z \frac{\mu^a(x, z)\mathbb{P}(\phi(Z) = \ell | Z = z)}{\mathbb{P}(A = a, \phi(Z) = \ell)} \mathbb{P}(A = a | Z = z)\mathbb{P}(Z = z)\, \mathrm{d}z \quad and \tag{2}$$

$$\pi_\phi(x, \ell) = \int_Z \frac{\pi(x, z)\mathbb{P}(\phi(Z) = \ell | Z = z)}{\mathbb{P}(\phi(Z) = \ell)} \mathbb{P}(Z = z)\, \mathrm{d}z. \tag{3}$$

*Then, under Assumptions 1, 2, and 3, the CATE $\tau(x)$ is bounded by*

$$b_\phi^-(x) \le \tau(x) \le b_\phi^+(x), \tag{4}$$

*with*

$$b_\phi^+(x) = \min_{l,m} b_{\phi;l,m}^+(x) \quad and \quad b_\phi^-(x) = \max_{l,m} b_{\phi;l,m}^-(x) \tag{5}$$

*where*

$$b_{\phi;l,m}^+(x) = \pi_\phi(x,l)\mu_\phi^1(x,l) + (1 - \pi_\phi(x,l))s_2 - (1 - \pi_\phi(x,m))\mu_\phi^0(x,m) - \pi_\phi(x,m)s_1, \tag{6}$$

$$b_{\phi;l,m}^-(x) = \pi_\phi(x,l)\mu_\phi^1(x,l) + (1 - \pi_\phi(x,l))s_1 - (1 - \pi_\phi(x,m))\mu_\phi^0(x,m) - \pi_\phi(x,m)s_2. \tag{7}$$

*Proof.* See Appendix A. □

Theorem 1 states that, in population, we yield valid closed-form bounds for $\tau(x)$ for arbitrary representations $\phi$. In particular, we can relax the optimization problem from Eq. (1) and obtain valid bounds $b_{\phi^*}{}^+(X) \ge b_*^+(X)$ and $b_{\phi^*}{}^-(X) \le b_*^-(X)$ by solving

$$\phi^* \in \underset{\phi \in \Phi}{\arg\min}\, \mathbb{E}_X[b_\phi{}^+(X) - b_\phi{}^-(X)]. \tag{8}$$

Here, we highlight the dependence of variables on the representation $\phi$ in green to show the differences to Eq. (1). Note the following differences: In contrast to Eq. (1), we do not impose any validity constraints in Eq. (8) because Theorem 1 automatically ensures the validity of our bounds. Furthermore, in contrast to Eq. (1), the objective from Eq. (8) only depends on identifiable quantities that can be estimated from observational data.

**Implications of Theorem 1:** A naïve implementation minimizing the bounds following Eq. (8) would require alternating learning. The reason is that, after every update step of $\phi(z)$, the quantities $\mu_\phi^a(x, l)$ and $\pi_\phi^a(x, l)$ are not valid for the updated $\phi$ anymore and would need to be retrained to ensure valid bounds. This is computationally highly expensive and causes unstable training as well as convergence problems. However, our method circumvents these issues: by using our novel Theorem 1, we show that, while training $\phi(z)$, the quantities $\mu_\phi^a(x, \ell)$ and $\pi_\phi(x, \ell)$ can be directly calculated. For that, we can simply evaluate the nuisance functions, which only need to be trained once in the first stage. This holds because our derivation of closed-forms bounds for arbitrary discrete representations of complex $Z$ comes with an important additional benefit: The bounds only depend on (i) discrete probabilities, (ii) quantities which are independent of $\phi$ and thus do not change for different $\phi$, and (iii) the discrete representation mapping to be learned itself. As a result, this allows us to *directly* learn $\phi$ wrt. Eq. (8). As such, we circumvent the need for adversarial or alternating training, which results in more robust estimation.

## 4.3 FINITE-SAMPLE VIEW

In practice, we have to estimate the bounds from Theorem 1 from finite observational data. For this purpose, we start with arbitrary initial estimators: $\hat{\pi}(x, z)$ is the estimator of the propensity score $\pi(x, z)$, $\hat{\mu}^a(x, z)$ of the response function $\mu^a(x, z)$, and $\hat{\eta}(z)$ of $\eta(z) = \mathbb{P}(A = 1 \mid Z = z)$.

Once the initial estimators are obtained, we can estimate our second-stage nuisance functions defined in Eq. (23) and (24) via

$$\hat{\mu}_\phi^a(x, \ell) = \frac{1}{\sum_{j=1}^n \mathbb{1}\{\phi(z_j) = \ell, a_j = a\}} \sum_{j=1}^n \hat{\mu}^a(x, z_j)\mathbb{1}\{\phi(z_j) = \ell\}(a\hat{\eta}(z_j) + (1 - a)(1 - \hat{\eta}(z_j))), \tag{9}$$

$$\hat{\pi}_\phi(x,\ell) = \frac{1}{\sum_{j=1}^n \mathbb{1}\{\phi(z_j) = \ell\}} \sum_j^n \hat{\pi}(x, z_j)\mathbb{1}\{\phi(z_j) = \ell\}. \tag{10}$$

Finally, we can directly 'plug in' these estimators into Eq. (5) to compute estimates of the upper and lower bound $\hat{b}_\phi^-(x), \hat{b}_\phi^+(x)$.

A naïve approach would now directly use $(\hat{b}_\phi^-(x), \hat{b}_\phi^+(x))$ to solve the optimization in Eq. (8). However, for finite samples, it turns out this is infeasible without restricting the complexity of the representation function. The reason is outlined in the following theoretical results.

**Lemma 1** (Tightness-bias-variance trade-off). *Let $\mathbb{E}_n$ and $\mathrm{Var}_n$ denote the expectation and variance with respect to the observational data (of size $n$). Then, it holds*

$$\mathbb{E}_n \Big[ \big( b_*^+(x) - \hat{b}_\phi^+(x) \big)^2 \Big] \le 2 \Big( \underbrace{\big( b_*^+(x) - b_\phi^+(x) \big)^2}_{\textit{(i) Population tightness}} + \underbrace{\mathbb{E}_n \big[ b_{\phi^*}^+(x) - \hat{b}_\phi^+(x) \big]^2}_{\textit{(ii) Estimation bias}} + \underbrace{\mathrm{Var}_n(\hat{b}_\phi^+(x))}_{\textit{(iii) Estimation variance}} \Big).$$
$$\tag{11}$$

*Proof.* See Appendix A. $\square$

**Interpretation of Lemma 1:** Lemma 1 shows that the mean squared error (MSE) between the estimated representation-based bound $\hat{b}_\phi^+(x)$ and the ground-truth optimal bound $b_*^+(x)$ can be decomposed into the following three components: (i) *population tightness*, (ii) *estimation bias*, and (iii) *estimation variance*. • Term (i) describes the discrepancy between the representation-based bound in population $b_\phi^+(x)$ and the ground-truth optimal bound $b_*^+(x)$. It will *decrease* if we allow for more complex representations $\Phi$, for example by increasing the number of partitions $k$. • Term (ii) describes the estimation bias due to using finite-sample estimators for estimating the bounds. It will generally depend on the type of estimators we employ for $\hat{\pi}(x,z)$, $\hat{\mu}^a(x,z)$, and $\hat{\eta}(z)$. • Finally, term (iii) characterizes the variance due to using finite-sample estimators. In contrast to term (i), it will *increase* when we allow the representation to be more complex. [3]

To make point (iii) more explicit, we derive the asymptotic distributions of the estimators from Eq. (9) and Eq. (10) that are used during training of $\phi$ to estimate the final bounds.

**Theorem 2** (Asymptotic distributions of estimators). *It holds that*

$$\sqrt{n}\hat{\mu}_\phi^a(x,\ell) \xrightarrow{d} \mathcal{N} \left( \mu_\phi^a(x,\ell), \frac{1}{p_{\ell,\phi}} \left( \frac{\mathrm{Var}(g(Z) \mid \phi(Z) = \ell)}{c} + d \right) \right) \tag{12}$$

$$\sqrt{n}\hat{\pi}_\phi(x,\ell) \xrightarrow{d} \mathcal{N} \left( \pi_\phi(x,\ell), \frac{1}{p_{\ell,\phi}} \mathrm{Var}(h(Z) \mid \phi(Z) = \ell) \right) \tag{13}$$

*for $c = q_{\ell,\phi}^2, d = \frac{\theta_\ell^2 (1 - p_{\ell,\phi} q_{\ell,\phi})}{q_{\ell,\phi}^3}$, such that $c, d > 0$ and where $p_{\ell,\phi} = \mathbb{P}(\phi(Z) = \ell)$, $q_{\ell,\phi} = \mathbb{P}(A = a \mid \phi(Z) = \ell)$, $g(Z) = \hat{\mu}^a(x, Z)(a\hat{\eta}(Z) + (1-a)(1-\hat{\eta}(Z)))$, $h(Z) = \hat{\pi}(x, Z)$, and $\theta_{\ell,\phi} = \mathbb{E}[g(Z) \mid \phi(Z) = \ell]$.*

*Proof.* See Appendix A. $\square$

We observe that the variance of the estimators (and, thus, of the estimated bounds) explodes for small values of $p_{\ell,\phi} = \mathbb{P}(\phi(Z) = \ell)$. Hence, to reduce the estimation variance, we aim to learn a representation $\phi$ that avoids low $p_{\ell,\phi}$ for some $\ell$, e.g., by limiting the number of partitions $k$. $\Rightarrow$ Altogether, as a consequence of Lemma 1 and Theorem 2, we obtain an *inherent trade-off between tightness of the bounds in populations and estimation variance in finite-samples*.

**Learning objective for the representation $\phi$:** Due to the inherent trade-off between tightness of the bounds and estimation variance, the aim for learning the representation $\phi$ is two-fold. On the one hand, we **(a)** aim to learn tight bounds, which is given in the objective in Eq. (8). On the other hand, we **(b)** also have to account for controlling the variance in finite-sample settings, especially for high-dimensional $Z$. Motivated by Theorem 2, we ensure $\hat{p}_{\ell,\phi} > \varepsilon$ for some $\varepsilon > 0$, where $\hat{p}_{\ell,\phi}$ is an estimator of $p_{\ell,\phi} = \mathbb{P}(\phi(Z) = \ell)$. Combining both **(a)** and **(b)** yields the following objective:

---

[3]Note that Lemma 1 and Theorem 2 hold for arbitrary $\phi$ and the corresponding bound estimators $\hat{b}_\phi^+(x)$. This allows us to ensure more stable update steps during training by reducing the estimation variance of the estimators. However, this implies that Lemma 1 and Theorem 2 and the following properties also directly hold for some finally learned or optimal $\phi^*$ which results in reduced variance of final estimates.

$$\phi^* \in \arg\min_{\phi \in \Phi} \mathbb{E}_X[\hat{b}_\phi^+(X) - \hat{b}_\phi^-(X)] \quad \text{s.t.} \quad \hat{p}_{\ell,\phi} > \varepsilon, \tag{14}$$

for some $\varepsilon > 0$ and all $\ell \in \{1, \ldots, k\}$.

Notably, the main motivation of Theorem 2 is *not* to construct confidence intervals or to provide theoretical results on the width of the finally learned bounds. Instead, we aim to yield valid final bound estimates by already ensuring valid bound estimates during training for robustly updating $\phi$. For that, we want to ensure that all nuisance functions are estimated with low variance at every update step to guarantee stable training. As a consequence, the final bounds built on top of these nuisance functions after training will also yield reliable estimates.

We next present a neural method to learn tight bounds using the above objective.

## 5 NEURAL METHOD FOR LEARNING CATE BOUNDS WITH COMPLEX INSTRUMENTS

In this section, we propose a neural method for our objective to learn tight and valid bounds. Our method consists of two separate stages (see Algorithm 1): ① we learn initial estimators of the three nuisance functions, and ② we learn an optimal representation $\phi^*$, so that the width of the bounds is minimized. Note that our method is completely model-agnostic. Hence, arbitrary machine learning models can be used in the first and second stages in order to account for the properties of the data. For example, for instruments with gene data, one could use pre-trained encoders to further optimize the downstream performance. We give an overview of the workflow of our method in Fig. 3 (see Algorithm 1 for pseudocode).

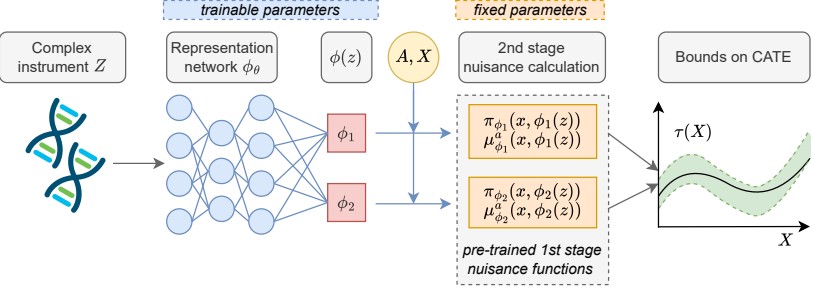

Figure 3: Workflow of the second stage of our method for calculating bounds on the CATE: The representation network $\phi_\theta$ learns discrete latent representations of the complex $Z$ (e.g., continuous or high-dimensional). By employing the pre-trained $\hat{\mu}$, $\hat{\pi}$, and $\hat{\eta}$, we can directly calculate the nuisance estimates conditional on the latent representation $\phi(z)$ by using Eq. (9) and Eq. (10) to yield the bounds.

① **Initial nuisance estimation:** In the first stage, we can use arbitrary machine learning models (e.g., feed-forward neural network) to learn the first-stage nuisance functions $\hat{\mu}^a(x, z) = \hat{\mathbb{E}}[Y \mid X = x, A = a, Z = z]$, $\hat{\pi}(x, z) = \hat{\mathbb{P}}(A = 1 \mid X = x, Z = z)$, and $\hat{\eta}(z) = \hat{\mathbb{P}}(A = 1 \mid Z = z)$.

Recall that we consider $Z$ and $X$, which are both potentially high-dimensional. Hence, for $\hat{\mu}^a(x, z)$ and $\hat{\pi}(x, z)$, we use network architectures that have (i) different encoding layers for $X$ and $Z$, so that we capture structured information within the variables and (ii) shared layers on top of the encoding to learn common structures. Further, for $\hat{\mu}^a(x, z)$, we use two outcome heads for both treatment options $A \in \{0, 1\}$ to ensure that the influence of the treatment on the outcome prediction does not 'get lost' in the high-dimensional space of $X$ and $Z$ (Shalit et al., 2017).

② **Representation learning:** In the second stage, we train a neural network to learn discrete representations of the instruments with the objective of obtaining tight bounds but with constraints on the estimation variance. To learn the function $\phi(z)$, we use a neural network $\phi_\theta$ with trainable parameters $\theta$. Then, on top of the final layer of the encoder, we leverage the Gumbel-softmax trick (Jang et al., 2017), which allows us to learn $k$ *discrete* representations of the latent space of the instruments, where $k$ can be flexibly chosen as a hyperparameter.

**Custom loss function:** We further transform our objective into a loss function to train the network $\phi_\theta$. For that, we design a compositional loss consisting of three terms:

①  A *bound-width minimization loss* that aims at our objective in Eq. (14), defined via

$$\mathcal{L}_{\mathrm{b}}(\theta) = \frac{1}{n} \sum_{i=1}^{n} \hat{b}_{\phi_\theta}^{+}(x_i) - \hat{b}_{\phi_\theta}^{-}(x_i) \tag{15}$$

②  A *regularization loss* to enforce the constraints in Eq. (14), i.e., enforcing that $\hat{p}_{\ell,\phi} = \hat{\mathbb{P}}(\phi_\theta(Z) = \ell) > \varepsilon, \forall \ell \in 1, \ldots, k$, for some $\varepsilon > 0$. For that, we aim to penalize the negative log-likelihood $-\sum_{j=1}^{k} \log(\mathbb{P}(\phi_\theta(Z) = j))$, which we can estimate via

$$\mathcal{L}_{\mathrm{reg}}(\theta) = -\sum_{j=1}^{k} \log\left(\frac{1}{n}\sum_{i=1}^{n} \mathbb{1}\{\phi_\theta(z_i) = j\}\right). \tag{16}$$

③  An *auxiliary guidance loss* $\mathcal{L}_{\mathrm{aux}}(\theta)$, which enforces more heterogeneity between $\mathbb{P}(Z \mid \phi_\theta(Z) = l)$ and $\mathbb{P}(Z \mid \phi_\theta(Z) = m)$, for all $l, m$. To achieve this, we add an additional linear classification head $p_\zeta$ with weights $\zeta$ on top of the last hidden layer of $\phi_\theta$ before the discretization. The auxiliary guidance loss is explicitly defined as the cross-entropy loss via

$$\mathcal{L}_{\mathrm{aux}}(\theta) = -\frac{1}{n} \sum_{i=1}^{n} \sum_{j=1}^{k} \mathbb{1}\{\phi_\theta(z_i) = j\} \log\left(p_\zeta(z_i)\right), \tag{17}$$

where $p_\zeta(z_i)$ is the predicted probability of assigning $z_i$ to discrete representation $j$ by the additional classification head. While $\mathcal{L}_{\mathrm{aux}}(\theta)$ is not strictly necessary for our objective, we empirically observed that it helps stabilize training by avoiding convergence to non-informative local minima. Hence, we yield our final training loss

$$\mathcal{L}(\theta) = \mathcal{L}_{\mathrm{b}}(\theta) + \lambda \mathcal{L}_{\mathrm{reg}}(\theta) + \gamma \mathcal{L}_{\mathrm{aux}}(\theta), \tag{18}$$

with hyperparameters $\lambda$ and $\gamma$. Here, $\lambda$ controls the trade-off between bound tightness and estimation variance, and can thus be tailored depending on the application. The hyperparameter $\gamma$ can be simply tuned as usual.

The main benefit of our method is that it is particularly efficient and robust compared to other learning procedures (such as alternating learning procedure or adversarial training). In the second stage, we solely update the parameters $\theta$ of the discretization network $\phi_\theta$ to minimize $\mathcal{L}_\theta$. In contrast, the networks of the first-stage nuisance estimators have frozen weights. In the second stage, networks of the first-stage nuisance estimators are only evaluated but are *not* updated. This allows us to re-use the trained first-stage networks for different training settings of the second-stage network (e.g., varying $k$). Thus, this results in a training procedure that is computationally more effective and robust.

**Robustness across $k$:** Our method is directly designed to be robust across the choice of the number of partitions $k$. This is due to its neural backbone and custom loss that encourages learning flexible representations that minimize the bound width already for low $k$ while ensuring robust estimation also for higher $k$ through regularization. This is particularly advantageous in real-world causal inference tasks, where model evaluation and selection are challenging due to the lack of oracle performance metrics. In the following experiments section, we demonstrate such robustness empirically. Further, we provide an extended discussion including practical guidelines in Appendix F.

## 6 EXPERIMENTS

**Baselines:** Existing methods (see Sec. 2) focus either on (a) point identification with strong assumptions, (b) partial identification with continuous treatment variables, or (c) discrete instruments. We instead focus on a setting with complex instruments and binary treatments. Hence, existing methods are **not** tailored to our setting, because of which a fair comparison is precluded. Instead, we thus demonstrate the validity and tightness of our bounds. Further, for comparison, we propose an additional NAÏVE baseline, which first learns a discretization of the instruments (via $k$-means clustering) and then learns the nuisance functions wrt. to the discretized instruments to apply the existing bounds for discrete instruments from Lemma 2 on top.

| Metric | Dataset 1 | | | Dataset 2 | | |
|---|---|---|---|---|---|---|
| | Naïve | Ours | Rel. Improvement | Naïve | Ours | Rel. Improvement |
| **Coverage[↑]** | $1.00 \pm 0.00$ | $1.00 \pm 0.00$ | 0.00% | $1.00 \pm 0.00$ | $1.00 \pm 0.00$ | 0.00% |
| **Width[↓]** | $1.22 \pm 0.05$ | $\mathbf{1.05 \pm 0.01}$ | 13.9% | $1.31 \pm 0.16$ | $\mathbf{1.14 \pm 0.16}$ | 13.0% |
| **MSD[↓]** | $0.28 \pm 0.06$ | $\mathbf{0.03 \pm 0.03}$ | 89.3% | $0.09 \pm 0.06$ | $\mathbf{0.06 \pm 0.06}$ | 33.3% |

Table 1: **Datasets 1 and 2**: Comparison of both methods (NAÏVE vs. Ours) regarding width, and MSD. Relative performance improvements in green.

**Data:** We perform experiments mimicking Mendelian Randomization but where we simulate the data to have access to the ground-truth CATE for performance evaluations, so that we can check for coverage and validity of the bounds. We consider three different realistic settings. For Datasets 1 and 2, we consider a one-dimensional continuous instrument representing a polygenic risk score (Pierce et al., 2018).
Further, in Dataset 1, we model the true $\pi(x, z)$ as a rather simple function to check if our method is already competitive in such settings. In Dataset 2, we model $\pi(x, z)$ as a complex function to evaluate the performance in more challenging settings. We use the same CATE for Dataset 1 and Dataset 2 to allow for comparisons between both. In Dataset 3, we model high-dimensional instruments

| Metric | Naïve | Ours | Rel. Improve |
|---|---|---|---|
| **Coverage (oracle)[↑]** | $0.96 \pm 0.09$ | $\mathbf{0.99 \pm 0.01}$ | 3.4% |
| **Width*[↓]** | $1.88 \pm 0.04$ | $\mathbf{1.85 \pm 0.04}$ | 1.8% |
| **MSE*[↓]** | $0.12 \pm 0.01$ | $\mathbf{0.11 \pm 0.01}$ | 9.2% |
| **MSD[↓]** | $0.10 \pm 0.10$ | $\mathbf{0.03 \pm 0.02}$ | 70.3% |

Table 2: **Dataset 3**: Comparison of both methods (NAÏVE vs. Ours) regarding the coverage with respect to the oracle bounds, width, and MSD. Relative performance improvements in green.

with single nucleotide polymorphisms (SNPs, i.e., genetic variants; Burgess et al., 2020) to test our method in an additional realistic and even more complex setting. In all datasets, we model the CATE to be heterogeneously conditioned on $X$ to check whether the bounds adapt to different subpopulations. Details are in Appendix D.

**Performance metrics:** We report the following metrics to assess the validity and robustness of the estimated bounds: (i) The *coverage*, i.e., how often the true CATE lies within the estimated bounds. (ii) The average *width* of bounds, where lower values indicate more informative bounds. (iii) The *mean squared difference* (*MSD*) of the predicted bounds over different values of $k$, indicating the robustness wrt. to the selection of the hyperparameter. Further, for Dataset 3, we model $\pi(x, z)$ to be dependent on some latent discrete representation of the observed $Z$, such that we can approximate oracle bounds. Thus,

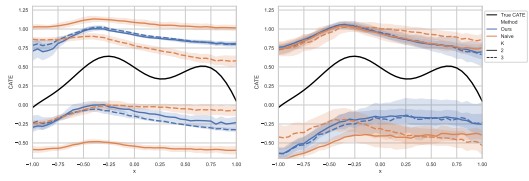

Figure 4: **Datasets 1 and 2: Estimated bounds on the CATE.** mean $\pm$ sd over 5 runs for different $k$. *Left*: Dataset 1 with a simple $\pi(x, z)$. *Right*: Dataset 2 with a complex $\pi(x, z)$.

we can evaluate (iv) the coverage wrt. to the oracle bounds and (v) the MSE to the oracle bounds. Recall that, for reliable decision-making, we would like to obtain tight bounds *but only under* the constraint that they yield valid coverage. We thus propose two new metrics, which we call *width\** and *MSE\**, which denote the corresponding metrics but where we filter for runs with coverage $\geq 95\%$. This allows us to properly compare the ability to learn tight bounds without distortions due to falsely overconfident predictions.

**Implementation details:** For our method, we use multi-layer-perceptrons (MLPs) for the first-stage nuisance estimation and an MLP with Gumbel-softmax (Jang et al., 2017) discretization on the last layer for learning $\phi_\theta$. For the NAÏVE baseline, we use $k$-means clustering in the first step to learn discretized instruments and then use MLPs with identical architecture for the nuisance estimation to ensure a fair comparison. We provide further details in Appendix C.

**Results:** We present the results of our experiments in Table 1 (for Datasets 1 and 2) and in Table 2 (for Dataset 3). Therein, we compare our method against the NAÏVE baseline averaged over multiple runs and over different choices of clusters $k$. Overall, we observe the following patterns: **(i)** Both methods (i.e., ours and the NAÏVE baseline) almost always reach a perfect coverage of 100% for the true CATE, which shows the validity of the bounds. For Dataset 3, our method achieves better coverage wrt. to the oracle bounds, which further suggests that our method leads to a more reliable

estimation. **(ii)** As expected, on average, our method learns *tighter bounds* for Datasets 1 and 2 (lower width), and for Dataset 3 our method learns tighter *valid* bounds that are closer to the oracle bounds (lower width* and MSE*). This demonstrates that our method can clearly improve over a discretization that uses solely information of $Z$ in the first step (NAÏVE). **(iii)** Unlike the baseline, our method is robust over different values of $k$. This is demonstrated by a low MSD in all datasets, with improvements up to 89% over the naïve baseline.

**Sensitivity over $k$:** To better understand the robustness as well as the source of performance gain of our method, we analyze the behavior of the methods for different parameters $k$. For that, we report the performance metrics for varying $k$ in Table 3 and Table 4. Also, we plot the estimated bounds for Datasets 1 and 2 in Fig. 4, and the estimated bound width over varying $k$ for Dataset 3 in Fig. 5. Overall, we observe robust behavior of our method but unstable behavior of the NAÏVE baseline wrt. $k$. The latter is also

| Dataset | Method | $k$ | Coverage[↑] | Width[↓] |
|---|---|---|---|---|
| Dataset 1 | Naïve | 2 | $1.00 \pm 0.00$ | $1.62 \pm 0.06$ |
| | | 3 | $1.00 \pm 0.00$ | $0.83 \pm 0.16$ |
| | Ours | 2 | $1.00 \pm 0.00$ | $1.01 \pm 0.05$ |
| | | 3 | $1.00 \pm 0.00$ | $1.09 \pm 0.04$ |
| Dataset 2 | Naïve | 2 | $1.00 \pm 0.00$ | $1.34 \pm 0.19$ |
| | | 3 | $1.00 \pm 0.00$ | $1.28 \pm 0.20$ |
| | Ours | 2 | $1.00 \pm 0.00$ | $1.13 \pm 0.19$ |
| | | 3 | $1.00 \pm 0.00$ | $1.15 \pm 0.31$ |

Table 3: **Datasets 1 and 2:** Comparison of methods across key metrics.

clearly visible by the large differences in the learned bounds in Fig. 4 on the left, and the high variation in estimated bound width in Fig. 5. This even results in learning falsely overconfident bounds for $k = 6$, as also shown by low oracle coverage in Table 4.

In contrast, our method yields bounds that are valid for a given $k$ as well as over varying values of $k$, which is naturally encouraged by our objective of flexibly learning representations. We thus see that our method is robust regardless of the parameter $k$, meaning that $k$ is not responsible for the performance gain but

| Method | $k$ | Coverage[↑] | Width[↓] | Coverage (oracle)[↑] | MSE (oracle)[↓] |
|---|---|---|---|---|---|
| Naïve | 2 | $1.00 \pm 0.00$ | $1.96 \pm 0.05$ | $1.00 \pm 0.00$ | $0.15 \pm 0.02$ |
| | 4 | $1.00 \pm 0.00$ | $1.91 \pm 0.03$ | $1.00 \pm 0.00$ | $0.13 \pm 0.02$ |
| | 6 | $1.00 \pm 0.00$ | $1.74 \pm 0.26$ | $0.75 \pm 0.50$ | $0.09 \pm 0.05$ |
| | 8 | $1.00 \pm 0.00$ | $1.89 \pm 0.09$ | $1.00 \pm 0.00$ | $0.12 \pm 0.04$ |
| Ours | 2 | $1.00 \pm 0.00$ | $1.87 \pm 0.05$ | $1.00 \pm 0.00$ | $0.12 \pm 0.02$ |
| | 4 | $1.00 \pm 0.00$ | $1.87 \pm 0.08$ | $1.00 \pm 0.00$ | $0.12 \pm 0.03$ |
| | 6 | $1.00 \pm 0.00$ | $1.85 \pm 0.06$ | $1.00 \pm 0.00$ | $0.11 \pm 0.02$ |
| | 8 | $1.00 \pm 0.00$ | $1.83 \pm 0.07$ | $0.99 \pm 0.01$ | $0.11 \pm 0.03$ |

Table 4: **Dataset 3 (high-dimensional)**: Comparison of methods across key metrics.

that this is due to our proposed learning objective. We provide an extended discussion about the role of $k$ and a practical guideline for selection in Appendix F.

**Takeaways:** Our method can successfully learn bounds that have a high coverage and a low width. Further, our method outperforms the NAÏVE baseline clearly while ensuring robustness. Here, our results show that the source of the performance gain is the way how we learn the representation $\phi$ and that the performance gain from our method becomes larger for more complex datasets.

**Limitations:** Our method for partial identification allows us to relax multiple assumptions that are inherent to methods for point identification. Nevertheless, we still rely on the standard assumptions of IV settings. However, such assumptions often hold by design or can be ensured by expert knowledge such as in Mendelian randomization. Appendix B. Further, we show the asymptotic behavior of the nuisance estimates to motivate our regularization loss for improving training stability. Deriving asymptotic properties on final bound estimators (e.g., to derive uncertainty estimates or estimators that are efficient or multiply robust) is thus a promising direction for future research in partial identification, not only for complex IVs but even for simple discrete settings.

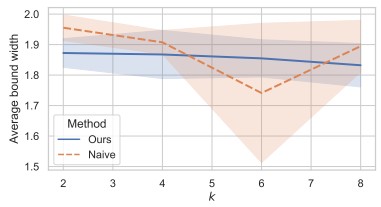

Figure 5: **Dataset 3 (high-dimensional):** Sensitivity analysis wrt. to the number of partitions $k$ where we show the average bound width $\pm$ sd over 5 runs.

We provide an extended discussion in

**Conclusion:** We propose a novel method for learning tight bounds on treatment effects by making use of complex instruments (e.g., instruments that are continuous, potentially high-dimensional, and that have non-trivial relationships with the treatment intake or exposure).

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

## A  PROOFS

### A.1  PROOF OF THEOREM 1

We begin by stating a result from the literature that obtains valid bounds for discrete instruments.

**Lemma 2** ((Swanson et al., 2018; Schweisthal et al., 2024)). *Under Assumptions 1 and 2, the CATE is bounded via*

$$b^-(x) \leq \tau(x) \leq b^+(x), \tag{19}$$

*with*

$$b^+(x) = \min_{l,m} b^+_{l,m}(x) \quad \text{and} \quad b^-(x) = \max_{l,m} b^-_{l,m}(x) \tag{20}$$

*where*

$$b^+_{l,m}(x) = \pi(x,l)\mu^1(x,l) + (1 - \pi(x,l))s_2 - (1 - \pi(x,m))\mu^0(x,m) - \pi(x,m)s_1, \tag{21}$$

$$b^-_{l,m}(x) = \pi(x,l)\mu^1(x,l) + (1 - \pi(x,l))s_1 - (1 - \pi(x,m))\mu^0(x,m) - \pi(x,m)s_2. \tag{22}$$

*Proof of Theorem 1.* First, note that, for a given representation $\phi$, the representation $\phi(Z)$ is still a valid (discrete) instrument that satisfies Assumptions 1 and 2. Hence, we can apply Lemma 2 using $\phi(Z)$ as an instrument and immediately obtain the bounds from Theorem 1, but with *representation-induced nuisance functions* $\mu^a_\phi(x, \ell) = \mathbb{E}[Y | X = x, A = a, \phi(Z) = \ell]$ and $\pi_\phi(x, \ell) = \mathbb{P}(A = 1 | X = x, \phi(Z) = \ell)$ for $\ell \in \{0, \dots, k\}$.

We can write the representation-induced response function as

$$\mathbb{E}[Y | X = x, A = a, \phi(Z) = \ell] \overset{Z \perp\!\!\!\perp X}{=} \int_Z \mathbb{E}[Y | X = x, A = a, Z = z]\mathbb{P}(Z = z | A = a, \phi(Z) = \ell)\, \mathrm{d}z$$

$$= \int_Z \mathbb{E}[Y | X = x, A = a, Z = z] \frac{\mathbb{P}(\phi(Z) = \ell | A = a, Z = z)\mathbb{P}(A = a | Z = z)\mathbb{P}(Z = z)}{\mathbb{P}(A = a | \phi(Z) = \ell)\mathbb{P}(\phi(Z) = \ell)}\, \mathrm{d}z$$

$$= \frac{1}{\mathbb{P}(A = a | \phi(Z) = \ell)\mathbb{P}(\phi(Z) = \ell)}$$

$$\int_Z \mathbb{E}[Y | X = x, A = a, Z = z]\mathbb{P}(\phi(Z) = \ell | A = a, Z = z)\mathbb{P}(A = a | Z = z)\mathbb{P}(Z = z)\, \mathrm{d}z$$

$$= \frac{1}{\mathbb{P}(A = a | \phi(Z) = \ell)\mathbb{P}(\phi(Z) = \ell)}$$

$$\int_Z \mathbb{E}[Y | X = x, A = a, Z = z]\mathbb{P}(\phi(Z) = \ell | Z = z)\mathbb{P}(A = a | Z = z)\mathbb{P}(Z = z)\, \mathrm{d}z \tag{23}$$

and the representation-induced propensity score as

$$\mathbb{P}(A = 1 | X = x, \phi(Z) = \ell) \overset{Z \perp\!\!\!\perp X}{=} \int_Z \mathbb{P}(A = 1 | X = x, Z = z)\mathbb{P}(Z = z | \phi(Z) = \ell)\, \mathrm{d}z$$

$$= \int_Z \mathbb{P}(A = 1 | X = x, Z = z)\mathbb{P}(\phi(Z) = \ell | Z = z)\frac{\mathbb{P}(Z = z)}{\mathbb{P}(\phi(Z) = \ell)}\, \mathrm{d}z \tag{24}$$

$$= \frac{1}{\mathbb{P}(\phi(Z) = \ell)} \int_Z \mathbb{P}(A = 1 | X = x, Z = z)\mathbb{P}(\phi(Z) = \ell | Z = z)\mathbb{P}(Z = z)\, \mathrm{d}z,$$

which completes the proof.  $\square$

## A.2 PROOF OF LEMMA 1

*Proof.* The result follows from

$$\mathbb{E}_n\left[\left(b_*^+(x) - \hat{b}_\phi^+(x)\right)^2\right] = \mathbb{E}_n\left[\left(b_*^+(x) - b_{\phi^*}^+(x) + b_{\phi^*}^+(x) - \hat{b}_\phi^+(x)\right)^2\right] \tag{25}$$

$$\leq 2\left(\left(b_*^+(x) - \hat{b}_\phi^+(x)\right)^2 + \mathbb{E}_n\left[\left(b_{\phi^*}^+(x) - \hat{b}_\phi^+(x)\right)^2\right]\right) \tag{26}$$

$$\overset{(*)}{=} 2\left(\left(b_*^+(x) - \hat{b}_\phi^+(x)\right)^2 + \mathbb{E}_n\left[b_{\phi^*}^+(x) - \hat{b}_\phi^+(x)\right]^2 + \mathrm{Var}_n(\hat{b}_\phi^+(x))\right), \tag{27}$$

where we used the bias-variance decomposition for the MSE for $(*)$. $\qquad\square$

## A.3 PROOF OF THEOREM 2

*Proof.* We derive the asymptotic distributions of the estimators $\hat{\mu}_\phi^a(x, \ell)$ from Eq. (9) and $\hat{\pi}_\phi(x, \ell)$ from Eq. (10). We proceed by analyzing the numerator and denominator of each estimator. First, we show that both are asymptotically normal and then we apply the delta method to obtain the asymptotic distribution of the ratios.

**Distribution of $\hat{\mu}_\phi^a(x, \ell)$:** Recall from Equation (9) that we can write $\hat{\mu}_\phi^a(x, \ell)$ as

$$\hat{\mu}_\phi^a(x, \ell) = \frac{S_n}{N_n}, \tag{28}$$

where

$$S_n = \frac{1}{n}\sum_{j=1}^n W_j, \quad \text{with} \quad W_j = \hat{\mu}^a(x, z_j)\mathbb{1}\{\phi(z_j) = \ell\}[a\hat{\eta}(z_j) + (1-a)(1 - \hat{\eta}(z_j))], \tag{29}$$

$$N_n = \frac{1}{n}\sum_{j=1}^n D_j, \quad \text{with} \quad D_j = \mathbb{1}\{\phi(z_j) = \ell, a_j = a\}. \tag{30}$$

We define the moments

$$\mu_W = \mathbb{E}[W] = p_\ell \theta_\ell \tag{31}$$

$$\sigma_W^2 = \mathrm{Var}(W) = p_\ell(\gamma_\ell - p_\ell \theta_\ell^2) \tag{32}$$

$$\mu_D = \mathbb{E}[D] = p_\ell q_\ell \tag{33}$$

$$\sigma_D^2 = \mathrm{Var}(D) = p_\ell q_\ell(1 - p_\ell q_\ell) \tag{34}$$

$$c_{WD} = \mathrm{Cov}(W, D) = p_\ell q_\ell \theta_\ell(1 - p_\ell), \tag{35}$$

where $p_\ell = \mathbb{P}(\phi(Z) = \ell)$, $q_\ell = \mathbb{P}(A = a \mid \phi(Z) = \ell)$, $\theta_\ell = \mathbb{E}[g(Z) \mid \phi(Z) = \ell]$, and $\gamma_\ell = \mathbb{E}[g(Z)^2 \mid \phi(Z) = \ell]$, with $g(Z) = \hat{\mu}^a(x, Z)(a\hat{\eta}(Z) + (1-a)(1 - \hat{\eta}(Z)))$. Note that, for better readability, in this proof we avoid the double indexing showing the dependency on $\phi$ which we used in the theorem in the main paper.

By the central limit theorem, we know that

$$\sqrt{n}\begin{pmatrix} S_n \\ N_n \end{pmatrix} \xrightarrow{d} \mathcal{N}_2\left(\mu = \begin{pmatrix} \mu_W \\ \mu_D \end{pmatrix}, \Sigma = \begin{pmatrix} \sigma_W^2 & c_{WD} \\ c_{WD} & \sigma_D^2 \end{pmatrix}\right). \tag{36}$$

Let $f(s, n) = \frac{s}{n}$. We are interested in the asymptotic distribution of the ratio $\hat{\mu}_\phi^a(x, \ell) = f(S_n, N_n)$. The delta method states that

$$\sqrt{n}f(S_n, N_n) \xrightarrow{d} \mathcal{N}_2\left(f(\mu_W, \mu_D), \nabla f^\top(\mu_W, \mu_D)\Sigma\nabla f(\mu_W, \mu_D)\right) \tag{37}$$

Using that the gradient is $\nabla f^\top(\mu_W, \mu_D) = \left( \dfrac{1}{\mu_D}, -\dfrac{\mu_W}{\mu_D^2} \right)$, we can obtain the asymptotic variance via

$$\nabla f^\top(\mu_W, \mu_D) \Sigma \nabla f(\mu_W, \mu_D) = \frac{\sigma_W^2}{\mu_D^2} - 2\frac{\mu_W c_{WD}}{\mu_D^3} + \frac{\mu_W^2 \sigma_D^2}{\mu_D^4} \tag{38}$$

$$= \frac{1}{p_\ell} \left( \frac{(\gamma_\ell - \theta_\ell^2)}{q_\ell^2} + \frac{\theta_\ell^2(1 - p_\ell q_\ell)}{q_\ell^3} \right) \tag{39}$$

$$= \frac{1}{p_\ell} \left( \frac{\mathrm{Var}(g(Z) \mid \phi(Z) = \ell)}{q_\ell^2} + \frac{\theta_\ell^2(1 - p_\ell q_\ell)}{q_\ell^3} \right). \tag{40}$$

**Distribution of $\hat{\pi}_\phi(x, \ell)$:** Recall from Equation (10) that we can write $\hat{\pi}_\phi(x, \ell)$ as

$$\hat{\pi}_\phi(x, \ell) = \frac{S_n}{N_n}, \tag{41}$$

where

$$S_n = \frac{1}{n} \sum_{j=1}^{n} W_j, \quad \text{with} \quad W_j = \hat{\pi}(x, z_j) \mathbb{1}\{\phi(z_j) = l\}, \tag{42}$$

$$N_n = \frac{1}{n} \sum_{j=1}^{n} D_j, \quad \text{with} \quad D_j = \mathbb{1}\{\phi(z_j) = l\}. \tag{43}$$

We define the moments

$$\mu_W = \mathbb{E}[W] = p_\ell \theta_\ell \tag{44}$$

$$\sigma_W^2 = \mathrm{Var}(W) = p_\ell(\gamma_\ell - p_\ell \theta_\ell^2) \tag{45}$$

$$\mu_D = \mathbb{E}[D] = p_\ell \tag{46}$$

$$\sigma_D^2 = \mathrm{Var}(D) = p_\ell(1 - p_\ell) \tag{47}$$

$$c_{WD} = \mathrm{Cov}(W, D) = p_\ell \theta_\ell(1 - p_\ell), \tag{48}$$

where $p_\ell = \mathbb{P}(\phi(Z) = \ell)$, $\theta_\ell = \mathbb{E}[h(Z) \mid \phi(Z) = \ell]$, and $\gamma_\ell = \mathbb{E}[h(Z)^2 \mid \phi(Z) = \ell]$, with $h(Z) = \hat{\pi}(x, Z)$.

By the central limit theorem, we know that

$$\sqrt{n} \begin{pmatrix} S_n \\ N_n \end{pmatrix} \xrightarrow{d} \mathcal{N}_2 \left( \mu = \begin{pmatrix} \mu_W \\ \mu_D \end{pmatrix}, \Sigma = \begin{pmatrix} \sigma_W^2 & c_{WD} \\ c_{WD} & \sigma_D^2 \end{pmatrix} \right). \tag{49}$$

We can then calculate the asymptotic variance using the delta method as above and obtain

$$\nabla f^\top(\mu_W, \mu_D) \Sigma \nabla f(\mu_W, \mu_D) = \frac{\sigma_W^2}{\mu_D^2} - 2\frac{\mu_W c_{WD}}{\mu_D^3} + \frac{\mu_W^2 \sigma_D^2}{\mu_D^4} \tag{50}$$

$$= \frac{1}{p_\ell}(\gamma_\ell - \theta_\ell^2) \tag{51}$$

$$= \frac{1}{p_\ell} \mathrm{Var}(h(Z) \mid \phi(Z) = \ell). \tag{52}$$

$\square$

# B  REAL-WORLD RELEVANCE AND VALIDITY OF ASSUMPTIONS

In this section, we elaborate on the real-world relevance of our considered setting and show that our assumptions often hold and are even weaker than the ones of existing approaches. For that, we draw upon two real-world settings.

## B.1  MENDELIAN RANDOMIZATION

Mendelian randomization (MR; the main motivational example from our paper) is a widely used method from biostatistics to estimate the causal effect of some treatment or exposure (such as alcohol consumption) on some outcome (such as cardiovascular diseases). We refer to Pierce et al. (2018) for an introduction to MR, which also shows that MR is widely used in medicine. For that, genetic variants (such as different single nucleotide polymorphisms, SNPs) are used as instruments where it is known that they only influence the exposure but not directly the outcome. Our method for partial identification with complex instruments is perfectly suited for this common real-world application. Depending on the use case, either a predefined genetic risk score (Burgess et al., 2020) as a continuous variable, or up to hundreds of SNPs are used simultaneously as IVs to strengthen the power of the analysis, resulting in high-dimensional instruments (Pierce et al., 2018).

**Validity of assumptions:** The IV assumptions used in our paper such as the exclusion and independence assumptions can be ensured by expert knowledge (e.g., given some observed confounder age ($X$), genetic variations (Z) do not affect age) or, in some cases, they can be even directly tested for (Glymour et al., 2012). In contrast, as explained in Sec. 2, existing methods for MR rely on additional hard assumptions on top such as the knowledge about the parametric form of the underlying data-generating process. Especially with such high-dimensional IVs, misspecification of these models may result in significantly biased effect estimates. In contrast, our method does not rely on any parametric assumption and also no additional assumptions compared to previous methods, thus enabling more reliable causal inferences in the real-world application of MR by using *strictly weaker* assumptions than existing work.

## B.2  INDIRECT EXPERIMENTS

With indirect experiments (IEs), we show that, in principle, our method is not constrained to medical applications but is also highly useful in various other domains. IEs are widely applied in various areas such as social sciences or public health to estimate causal effects in settings with non-adherence, i.e., where people cannot be forced to take treatments but rather be encouraged by some nudge (Pearl, 1995). For instance, researchers might be interested in estimating the effect of some treatment such as participating in a healthcare program ($T$) on some health outcome $Y$ by randomly assigning nudges $Z$ (IVs) in the form of different text messages on social media promoting participation. Here, common nudges (IVs) are in the form of, for instance, text or even image data and thus high-dimensional, showing the necessity of a method capable of handling complex IVs such as ours.

In principle, our method can be applied to every setting with continuous or multi-dimensional IVs where one wants to avoid making the hard untestable assumptions necessary for point identification such as linearity or additivity (e.g., Hartford et al. (2017)). Specific examples for applications with high-dimensional IVs are text-based nudges for encouraging vaccinations (Milkman et al., 2021), or various kinds of experiments where text nudges are generated by different strategies such as for political microtargeting (Hackenburg & Margetts, 2024) or for personalized persuasion in general (Matz et al., 2024).

Another important application area is online marketing. Concrete use cases involve extended A/B testing for evaluating the benefits of new features, e.g., when one is interested in the effect of a new version of an app on user engagement. Here, users with features such as age, gender, and content preferences ($X$) can be nudged by emails or push notifications ($Z$) to test a new feature such as using a new version of an app ($A$) to estimate its effect on engagement metrics such as screen time ($Y$). Further, our method could also be extended to improve current methods for optimizing instrument designs for indirect experiments that for now assume identifiability is possible (e.g., Chandak et al. (2023)).

**Validity of assumptions:** As a major benefit of IEs, the IV assumptions are *ensured per design* as the IVs are randomly assigned, and, thus they always hold. Hence, our method provides a promising tool for evaluating the effects of IEs.

# C   IMPLEMENTATION AND TRAINING DETAILS

**Model architecture:** For all our models, we use MLPs with ReLU activation function. For $\hat{\mu}_\phi^a$, we use 2 layers to encode $X$ and 3 layers to encode $Z$. Then, we concatenate the outputs and add 2 additional shared layers. Finally, we calculate the outputs by a separate treatment head for $A = 0$ and $A = 1$ to ensure the expressiveness of $A$ for predicting $Y$. For $\hat{\pi}$, we use the same architecture. For $\hat{\eta}$, we use 3 layers. For $\phi_\theta$, we also use 3 layers and apply discretization on top of the $K$ outputs (Jang et al., 2017). For the nuisance parameters of the $k$-means baseline, we use the same models as for $\hat{\mu}_\phi^a$ and $\hat{\pi}$ for a fair comparison. We use a neuron size of 10 for all hidden layers.

**Training details:** For training our nuisance functions, we use an MSE loss for the functions learning the continuous outcome $Y$ and a cross-entropy loss for functions learning the binary treatment $A$. For all models, we use the Adam optimizer with a learning rate of $0.03$. We train our models for a maximum of 100 epochs and apply early stopping. For our method, we fixed $\lambda = 1$ and performed random search to tune for $[0, 1]$ for $\gamma$. We use PyTorch Lightning for implementation. Each training run of the experiments could be performed on a CPU with 8 cores in under 15 minutes.

## D  DATA DESCRIPTION

**Dataset 1:** We simulate an observed confounder $X \sim \text{Uniform}[-1,1]$ and an unobserved confounder $U \sim \text{Uniform}[-1,1]$.

The instrument $Z$ is defined as

$$Z \sim \text{Mixture}\left(\frac{1}{2}\text{Uniform}[-1,1] + \frac{1}{4}\text{Beta}(2,2) + \frac{1}{4}(-\text{Beta}(2,2))\right). \tag{53}$$

We define $\rho$ as

$$\rho = \frac{1}{1 + \exp\left(-\left((2|Z| - \max(Z)) + X + 0.5 \cdot U\right)\right)}. \tag{54}$$

Then, the propensity score is given by

$$\pi = (\rho - 0.5) \cdot 0.9 + 0.5. \tag{55}$$

We then sample our treatment assignments from the propensity scores as

$$A \sim \text{Bernoulli}(\pi). \tag{56}$$

The conditional average treatment effect (CATE) is defined as

$$\tau(X) = -\frac{(2.5X)^4 + 12\sin(6X) + 0.5\cos(X)}{80} + 0.5. \tag{57}$$

The outcome $Y$ is then generated by

$$Y = (X + 0.5U + 0.1 \cdot \text{Laplace}(0,1)) \cdot 0.25 + \tau(X) \cdot A. \tag{58}$$

**Dataset 2:** We keep the other properties but change the propensity score to be more complex, which results in harder-to-learn optimal representations of $Z$ for tightening the bounds. The propensity score is given by

$$\pi = \sin(2.5Z + X + U) \cdot 0.48 + 0.48 + \frac{0.04}{1 + \exp(-3|Z|)}. \tag{59}$$

**Dataset 3:** We simulate $X$ and $U$ as above. Then, we sample a $d$-dimensional $Z \in \{0,1\}^d$ with $d = 20$ as

$$Z \sim \text{Binomial}(d, 0.5). \tag{60}$$

Thus, our modeling is here inspired by using multiple SNPs (appearances of genetic variations) as instruments (Burgess et al., 2020), where we simulate potential variations for 20 genes.

Then, we define

$$\rho = \sum_{j=1}^{d}[\mathbb{1}\{j \leq 5\}Z_j] \tag{61}$$

and the propensity score, inspired by the more complex setting of Dataset 2, as

$$\pi = 0.48\sin(10\rho + X + U) + 0.48 + \frac{0.04}{1 + \exp(-3|5\rho|)}. \tag{62}$$

Then, we define the CATE as

$$\tau(X) = -\frac{-(1.6X + 0.5)^4 + 12\sin(4X + 1.5) + \cos(X)}{80} + 0.5. \tag{63}$$

and the outcome dependent on $\tau$, $X$ and $U$ analogously as for Datasets 1 and 2.

**Dataset 4:** To test our method even in higher-dimensional settings, we consider a 4th dataset with **100-dimensional IVs**. For that, we adapt the DGP from dataset 3 but set $d = 100$. Then we adjust the latent discrete IV score as

$$\rho = \sum_{j=1}^{d}[\mathbb{1}\{j \leq 25\}Z_j]. \tag{64}$$

By Eq. (61) and Eq. (64), we ensure that some of the modeled SNPs are irrelevant for $\pi$ and thus do not affect the treatment or exposure $A$. Thereby, we focus on realistic settings in practice, where the relevance of instruments cannot always be ensured which imposes challenges especially for existing methods for point identification, but not for our approach. Further, we ensure that the latent score $\rho$ can only take 5 discrete levels for dataset 3 and 25 discrete levels for dataset 4. This allows us to approximate oracle bounds using the discrete bounds on top of $\rho$ by leveraging Lemma 2 such that we can evaluate our method and the baseline in comparison to oracle bounds.

To create the simulated data used in Sec. 6, we sample $n = 2000$ from the data-generating process above. We then split the data into train (40%), val (20%), and test (40%) sets such that the bounds and deviation can be calculated on the same amount of data for training and testing.

# E  ADDITIONAL RESULTS

## E.1  ADDITIONAL BASELINES

As mentioned in the main paper, existing methods are not designed for our considered setting of continuous or high-dimensional IVs with binary treatments. However, to further show the advantages and necessity of our tailored method, we compare with two additional baselines that were not developed for our task but which we adapted for our task, namely, one from uncertainty quantification for point estimates and one from the discrete instruments setting:

(i) *DeepIV with bootstrapped confidence intervals*. DeepIV (Hartford et al., 2017) is a neural method tailored for high-dimensional instruments when point identification can be ensured. This requires the *additional assumption* of additivity of the unobserved confounding, which usually cannot be ensured and is not necessary for our method. For DeepIV, we can approximate confidence intervals using bootstrapping. Here, we approximate confidence intervals with a confidence level of $95\%$, indicating an expected coverage of $95\%$ if assumptions were not violated. However, note that these intervals can *only* adjust for statistical uncertainty, but *not* for identifiability uncertainty due to the violation of causal assumptions. Thus, this baseline acts as an additional motivation for why bound estimators such as our method are important.

(ii) *Discretized IVs*: As a further additional baseline, we proceed by directly discretizing the high-dimensional IVs and then estimating the existing bounds for discrete IVs. Hence, *one looses information* from the IV due to the discretization. Our implementation here is the same as for the naïve baseline, however, the $k$ partitions are not learned by $k$-means clustering but instead defined by a simple grouping rule. To ensure a fair comparison, we average the results of experiments conducted with the same number of partitions $k$ for all methods.

| Metric | DeepIV (CI) | Discretized | Naïve | Ours | Rel. Improvement |
|---|---|---|---|---|---|
| **Coverage[↑]** | $0.52 \pm 0.29$ | $1.00 \pm 0.00$ | $1.00 \pm 0.00$ | $1.00 \pm 0.00$ | 0.0% |
| **Coverage (oracle)[↑]** | $0.00 \pm 0.00$ | $0.99 \pm 0.01$ | $0.96 \pm 0.09$ | $\mathbf{0.99 \pm 0.01}$ | 0.0% |
| **Width*[↓]** | — | $1.91 \pm 0.04$ | $1.88 \pm 0.04$ | $\mathbf{1.85 \pm 0.04}$ | 1.8% |
| **MSE*[↓]** | — | $0.13 \pm 0.01$ | $0.12 \pm 0.01$ | $\mathbf{0.11 \pm 0.01}$ | 9.2% |
| **MSD[↓]** | — | $0.08 \pm 0.03$ | $0.10 \pm 0.10$ | $\mathbf{0.03 \pm 0.02}$ | 70.3% |

Table 5: **Dataset 3**: Comparison of methods (Naïve vs Ours) on coverage and width metrics with relative performance improvement. Note: "—" means that there are no reliable runs for which the corresponding performance metrics could be calculated.

**Results:** We report our results for Dataset 3 in Table 5. We observe that the DeepIV method, as expected, gives *falsely* overconfident bounds with only about $53\%$ coverage of the true CATE and no coverage of the oracle bounds. Thus, there are no reliable runs for which the other metrics could be calculated (denoted by "—" in the tables). This emphasizes the necessity for using bound estimators. Further, we observe that the discretized baseline gives *more conservative* and *wider* bounds under similar coverage (higher Width* and MSE*) and performs less robustly with regard to $k$ (higher MSD). In sum, the results confirm the strong performance of our method.

## E.2  HIGH-DIMENSIONAL DATASET

| Metric | DeepIV (CI) | Discretized | Naïve | Ours | Rel. Improvement |
|---|---|---|---|---|---|
| **Coverage[↑]** | $0.01 \pm 0.00$ | $1.00 \pm 0.00$ | $1.00 \pm 0.00$ | $1.00 \pm 0.00$ | 0.0% |
| **Coverage (oracle)[↑]** | $0.00 \pm 0.00$ | $1.00 \pm 0.00$ | $1.00 \pm 0.00$ | $1.00 \pm 0.00$ | 0.0% |
| **Width*[↓]** | — | $1.90 \pm 0.06$ | $1.82 \pm 0.13$ | $\mathbf{1.75 \pm 0.08}$ | 3.7% |
| **MSE*[↓]** | — | $0.26 \pm 0.03$ | $0.23 \pm 0.05$ | $\mathbf{0.21 \pm 0.03}$ | 10.9% |
| **MSD[↓]** | — | $0.05 \pm 0.03$ | $0.10 \pm 0.04$ | $\mathbf{0.05 \pm 0.01}$ | 48.2% |

Table 6: **Dataset 4** (100-dimensional IVs): Comparison of methods (Naïve vs Ours) on coverage and width metrics with relative performance improvement. Note: "—" means that there are no reliable runs for which the corresponding performance metrics could be calculated.

To show the validity of our method in even more high-dimensional settings, we added additional experiments with 100-dimensional IVs. For that, we introduced our Dataset 4 (see Appendix D). We report the results for our method and the same baselines as in the previous section. Further, for

the higher-dimensional setting, we varied the hyperparameter $k$ over $[2, 5, 7, 10, 20]$ for all bound estimation methods. We observe similar patterns as for our other dataset. In particular, the DeepIV baseline fails *entirely* to provide reliable bounds. In summary, our method shows robust performance by providing tighter and more reliable bounds than the baseline, even in high-dimensional settings. This emphasizes the applicability of our bounds in even more complex settings.

### E.3 ABLATION STUDYS

To further examine the robustness of our method in non-standard settings, we perform two additional ablation studies, one for varying the DGP and one for varying the selected nuisance models.

**Linear DGP with flexible models:** To analyze if our flexible method also performs robustly in simple settings, we evaluate our method which uses neural networks at every stage on a simple linear DGP. For that we adapt our Dataset 3 and use linear functions for the dependencies between the variables. We report the results in Table 7. As expected, our method performs also robustly in the simpler linear setting and outperforms the baseline by a clear margin again. Summarized, our method shows strong performance which emphasizes its applicability to datasets of various complexity levels.

| Metric | Naïve | Ours | Rel. Improve |
|---|---|---|---|
| **Coverage[↑]** | $1.00 \pm 0.00$ | $\mathbf{1.00 \pm 0.00}$ | 0.0 |
| **Coverage (oracle)[↑]** | $0.92 \pm 0.18$ | $\mathbf{1.00 \pm 0.00}$ | 8.6% |
| **Width*[↓]** | $2.07 \pm 0.04$ | $\mathbf{1.99 \pm 0.05}$ | 3.9% |
| **MSE*[↓]** | $0.10 \pm 0.01$ | $\mathbf{0.08 \pm 0.01}$ | 20.0% |
| **MSD[↓]** | $0.08 \pm 0.08$ | $\mathbf{0.04 \pm 0.03}$ | 50.0% |

Table 7: **Linear DGP**: Comparison of methods across key metrics. Relative performance improvements in green.

**Non-linear DGP with linear models:** In our method, we leverage neural networks at all stages to allow for consistent and flexible estimation of all properties. However, since our method is model-agnostic in principle, we analyze the behavior of our method when using non-flexible (mis-specified) models. For that, we implement our method and the baseline by using linear models for the nuisance estimates and evaluate the performance on our non-linear Dataset 3 (i.e., the nuisances and the bounds are misspecified). We report the results in Table 8. As expected, because of the misspecification of the nuisance models, full coverage of the bounds cannot be guaranteed. However, our method still outperforms the naive baseline evidently with respect to coverage and MSD while yielding similar bound tightness. Further, with coverage to the oracle bounds over $90\%$ and low MSD, our method still predicts close to valid bounds robustly over different runs which is unlike the naive baseline. This shows that our method is also robust against misspecification of the nuisance models as when using linear models for non-linear datasets.

| Metric | Naïve | Ours | Rel. Improve |
|---|---|---|---|
| **Coverage[↑]** | $0.96 \pm 0.06$ | $\mathbf{1.00 \pm 0.00}$ | 4.1% |
| **Coverage (oracle)[↑]** | $0.59 \pm 0.28$ | $\mathbf{0.91 \pm 0.04}$ | 54.2% |
| **Width*[↓]** | $1.91 \pm 0.02$ | $\mathbf{1.91 \pm 0.03}$ | 0.0% |
| **MSE*[↓]** | $0.14 \pm 0.04$ | $\mathbf{0.14 \pm 0.02}$ | 0.0% |
| **MSD[↓]** | $0.20 \pm 0.11$ | $\mathbf{0.02 \pm 0.01}$ | 90.0% |

Table 8: **Non-linear DGP with linear nuisance models**: Comparison of methods across key metrics. Relative performance improvements in green.

# F  ROLE OF NUMBER OF PARTITIONS $k$

## F.1  WHY OUR METHOD IS ROBUST TO DIFFERENT CHOICE OF $k$

One major advantage of our method is that it is clearly less sensitive to the hyperparameter $k$ than, for example, the naïve baseline. Empirically, we demonstrate this in our experiments by lower variance and stable behavior over varying $k$, especially visible in the low values of MSD. This is due to the combination of learning flexible representations tailored to minimize bound width (allowing us to estimate tight bounds already for low $k$) while ensuring reliable estimates of the nuisance functions in the second stage by using our regularization loss in Eq. (16) (ensuring robust behavior also for higher $k$).

Note that the robustness of our method is especially beneficial when applying our method to real-world settings in causal inference. In real-world settings from causal inference, hyperparameter tuning and model evaluation are not directly possible because oracle CATE or oracle bounds are not known. Thus, the robustness against suboptimal selection of hyperparameters such as $k$ is crucial. In the following, we provide further high-level theoretical insights into the role of $k$ and propose practical recommendations for selecting $k$ in real-world applications.

**Estimation error for different** $k$: The hyperparameter $\lambda$ controls the regularization loss in Eq. (16), i.e., it tries to maximize $\hat{p}_{\ell,\phi} = \hat{\mathbb{P}}(\phi_\theta(Z) = \ell) > \varepsilon$ for all $\ell \in 1, \ldots, k$. Thus, if we choose $\lambda$ high enough, then we enforce that $\hat{p}_{\ell,\phi} = 1/k$ for all $\ell \in 1, \ldots, k$. Plugged into Theorem 12, the asymptotic variances for the nuisance estimators are $k \left( \frac{\text{Var}(g(Z) \mid \phi(Z)=\ell)}{c} + d \right)$ for $\hat{\mu}^a_\phi(x, \ell)$, and $k \left( \text{Var}(h(Z) \mid \phi(Z) = \ell) \right)$ for $\hat{\pi}_\phi(x, \ell)$, respectively. Thus, for large enough $\lambda$, the variance of the nuisance estimators (and, thus, also likely of the final bounds) will increase for increasing $k$. However, as an interesting side note, for a fixed (not too large) $\lambda$, the penalization term in Eq. (16) will also grow with growing $k$ due to the same reason, which yields an automated stabilization for higher $k$. This is also shown in our experiments where higher values of $k$ do *not* necessarily result in a higher variance.

**Bound tightness for different** $k$: On a population level, the bounds get tighter with growing $k$. This follows straightforwardly from Theorem 1, since using more $k$ increases the flexibility of $\phi$. While the exact bound width is highly non-trivial, we can use results from Schweisthal et al. (2024) about bounds for the CATE with discrete instruments to give some intuition. Specifically, in our setting, for some $x$, the bound width is bounded by $b^+_\phi(x) - b^-_\phi(x) \leq \min_{l,m} \{(s_2 - s_1)(2 - \pi_\phi(x, \ell) - (1 - \pi_\phi(x, m)))\}$ with $l, m \in \{1, \ldots, k\}$. This has two major implications. First, if for some $x$, $\phi$ is learned such that $\phi(x, \ell)$ is close to 1 for some $l$ and $\pi_\phi(x, m)$ is close to 0 for some $m$, the bound width is close to zero ("point identification"). Second, if the optimal partitioning function $\phi$ is the same for all $x$ (implying $b(x) = b$), then setting $k = 3$ can be sufficient to yield the tightest bounds. This is because, by using a flexible network for $\phi$, the partitions can be learned such that partition 1 yields propensity scores as close as possible to zero (as the data allows), partition 2 yields propensity scores as close as possible to 1, and partition 3 contains all $z$ resulting in propensity scores between those values. Note, however, that this is only valid in population but can result in highly unreliable estimation in finite sample data.

## F.2  PRACTICAL GUIDELINES FOR SELECTING $k$

Although we showed that our method is designed to be robust against different selections of $k$, we provide two potential guidelines for how to choose $k$ in real-world settings where ground-truth CATE or bounds are not available for model selection.

*Approach 1: Expert-informed approach.* In some medical applications, physicians might already know or make an educated guess about a number of underlying clusters of patient characteristics such as genetic variants. For instance, this is a common assumption in subgroup identification or latent class analysis in medicine where patient groups are characterized by having similar responses to treatments or showing similar associations with diseases (Kongsted & Nielsen, 2017). Thus, no data-driven approach is necessary here but one can integrate existing domain knowledge.

*Approach 2: Data-driven for hypothesis confirmation.* Often, physicians are interested in whether some treatment or exposure has a positive or negative effect (i.e., lower bound $> 0$ or upper bound

$< 0$) for at least some observations $x$. Thus, $k$ can be selected by increasing $k$ until such an effect can be observed while holding the variance minimal. Then, the variance can be approximated (e.g., by bootstrapping to test for the reliability of the corresponding bound model and its effect). Thus, this approach can be used when our method is used as a support tool for hypothesis confirmation.

Last, straightforwardly, from an exploratory perspective, all hyperparameters ($k$, $\lambda$, $\gamma$) can be altered together to examine the behavior of bound width and estimation variance to post-hoc find a suitable hyperparameter configuration for a dataset that fulfills the subjective preferences of the practitioner.

# G SENSITIVITY ANALYSIS

We perform a sensitivity analysis over the hyperparameters in our custom loss function. We report the results in Fig. 6 and Fig. 7 for dataset 3 and for $k = 3$. We observe that $\gamma$ does not affect the bound size but can be optimized to reduce estimation variance, as mentioned in the motivation of our auxiliary guidance loss. Thus, $\lambda$ demonstrates the trade-off between tightness and variance and shows the importance of our regularization loss. Here, $\lambda$ can be increased to reduce the variance. In our experiments, the optimal trade-off between reduced variance and bound tightness also results in optimal oracle coverage, showing the practicability of our regularization.

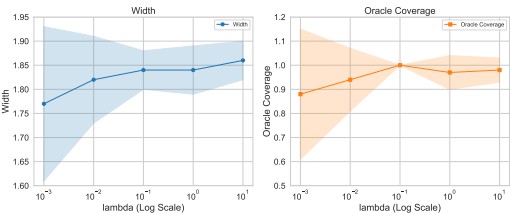

Figure 6: Sensitivity over $\lambda$. Left: Average bound width. Right: Oracle coverage. Averaged over 5 runs $\pm$ sd.

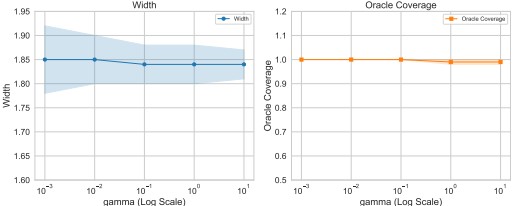

Figure 7: Sensitivity over $\gamma$. Left: Average bound width. Right: Oracle coverage. Averaged over 5 runs $\pm$ sd.

# H  Training procedure

---

**Algorithm 1:** Two-stage learner for estimating bounds with complex instruments

---

**Input** : observational data sampled from $(Z, X, A, Y)$, epochs $e$, batch size $n_b$, neural network $\phi_\theta$ with parameters $\theta$, learning rate $\delta$

**Output** : bounds $\hat{b}^-_{\phi_\theta}(x), \hat{b}^+_{\phi_\theta}(x)$

// First stage (nuisance estimation)

$\hat{\mu}^a(x, z) \leftarrow \hat{\mathbb{E}}[Y \mid X = x, A = a, Z = z]$

$\hat{\pi}(x, z) \leftarrow \hat{\mathbb{P}}(A = 1 \mid X = x, Z = z)$

$\hat{\eta}(z) \leftarrow \hat{\mathbb{P}}(A = 1 \mid Z = z)$

// Second-stage (partition learning and bound calculation)

**for** $\epsilon \in \{1, \ldots, e\}$ *in batches* **do**

    **for** $\ell \in \{1, \ldots, k\}$ **do**

        $\hat{\mu}^a_{\phi_\theta}(x, \ell) = \frac{1}{\sum_j^{n_b} \mathbb{1}\{\phi_\theta(z_j) = \ell, A = a\}} \sum_j^{n_b} \hat{\mu}^a(x, z_j)\mathbb{1}\{\phi_\theta(z_j) = \ell\}(a\hat{\eta}(z_j) + (1 - a)(1 - \hat{\eta}(z_j)))$

        $\hat{\pi}_{\phi_\theta}(x, \ell) = \frac{1}{\sum_j^{n_b} \mathbb{1}\{\phi_\theta(z_j) = \ell\}} \sum_j^{n_b} \hat{\pi}(x, z_j)\mathbb{1}\{\phi_\theta(z_j) = \ell\})$

    **end**

    $\hat{b}^+_{\phi_\theta}(x) = \min_{l,m} \hat{b}^+_{\phi_\theta; l,m}(x), \quad \hat{b}^-_{\phi_\theta}(x) = \max_{l,m} \hat{b}^-_{\phi_\theta; l,m}(x)$ for $l, m \in \{1, \ldots, K\}$

    $\mathcal{L}(\theta) \leftarrow \mathcal{L}_{\mathrm{b}}(\theta) + \lambda\mathcal{L}_{\mathrm{reg}}(\theta) + \gamma\mathcal{L}_{\mathrm{aux}}(\theta)$ as per Sec. 5

    $\theta \leftarrow \theta - \delta\nabla_\theta \mathcal{L}(\theta)$

**end**

// Final bounds

**return** $\hat{b}^-_{\phi_\theta}(x), \hat{b}^+_{\phi_\theta}(x)$

---

# I  DISCUSSION: DOUBLY ROBUSTNESS

**Background doubly robustness:** A related literature stream addressing robustness in causal inference aims to construct such called *doubly robust* or *multiply robust* estimators of causal quantities (see e.g., Bang & Robins (2005); Kennedy (2023)). Here, doubly / multiply robust means that the final estimator of the causal quantity (e.g., the ATE or CATE) is consistent if some of the nuisance estimators are consistent. For instance, under identifiability assumptions, the DR-learner for estimating the CATE (Kennedy, 2023) is consistent if either the outcome estimator $\hat{\mathbb{E}}[Y|X, A = a]$ or the propensity estimator $\hat{\mathbb{P}}(A = 1|X)$ is consistent. Other works extend such an idea for multiply robustness with additional nuisance estimators for other settings such as with IVs (Ogburn et al., 2015; Frauen & Feuerriegel, 2023). However, these methods consider only settings where the causal quantity can be point-identified, i.e., they require hard assumptions and are not tailored for estimating bounds, which is unlike our setting.

Only recently, doubly robust estimators have been proposed for bounds for the CATE (i.e., partial identification) for sensitivity analysis (Oprescu et al., 2023), and, closest to our setting, when IVs are available (Schweisthal et al., 2024). However, these bounds are only applicable for *discrete* IVs, which is unlike our setting with continuous or high-dimensional IVs.

*Why is our method not doubly robust?:* To derive doubly or multiply robust estimators, we would need to derive the efficient influence function of the causal quantity we want to estimate (Kennedy, 2023), i.e., in our setting the bounds for the CATE. However, under our assumptions 1-3, *no closed-form solution exists* for the bounds for the CATE for general IVs (i.e., continuous or high-dimensional). Instead, we can only describe the identification of the bounds as a constraint optimization problem (Gunsilius, 2020; Kilbertus et al., 2020) as we do in Eq (1). Since the constrained optimization problem is not pathwise-differentiable, the current statistical efficiency theory used for deriving doubly robustness is not applicable. Thus, deriving doubly robust estimators without a closed-form solution is not solvable with the usual toolkit and highly non-trivial. Instead, in a more general setting, related works try to solve such constrained optimization with different optimization methods such as alternating learning (e.g., Padh et al. (2023)), *and are also not doubly robust.*

*Potential extension with doubly robust estimation:* As stated above, we cannot directly derive doubly robust estimators for the bounds in our setting. However, as an advantage of our method, we try to learn optimal partitions (i.e., discretizations) of the IVs to yield reliable and tight bound estimates. This implies that after we finally learned our optimal partitions, we could replace the calculation of the final bounds for which we used Eq. (9) and Eq. (10), with an additional estimation procedure for discrete IVs such as by using the meta-learners for bounds of Schweisthal et al. (2024), including their doubly-robust learner. Note, however, that this only results in doubly robust estimates of the bounds on top of the learned partition but *not for the original problem.* Further, as this requires learning additional nuisance functions and does not use our optimized nuisance estimates, such a procedure might easily result in higher variance again and produces computational overhead. Therefore, we would not recommend this extension.

