# OpenReview forum: "Learning Representations of Instruments for Partial Identification of Treatment Effects"
_ICLR.cc/2025/Conference — Submitted to ICLR 2025_

### Official Review · Reviewer_ePQ5 · 2024-10-30

**Soundness:** 2
**Presentation:** 2
**Contribution:** 1
**Rating:** 3
**Confidence:** 4

**Summary:**

This paper introduces a method for estimating treatment effects from observational data despite violations of unconfoundedness. It presents a two-step procedure using high-dimensional latent instruments to derive valid bounds on the Conditional Average Treatment Effect.

**Strengths:**

1.	The study addresses an intriguing and valuable problem in causal inference.

2.	The writing is clear and accessible, making complex concepts easier to understand.

**Weaknesses:**

1. The method appears to be a straightforward combination of existing techniques, lacking clear theoretical innovation.

2. Although the paper highlights the advantage of utilizing high-dimensional instruments, there are limited experiments involving such instruments in the evaluation phase.

3. The experimental results show no substantial improvements, casting doubt on the method's effectiveness.

4. While the authors discuss the lack of comparison implementations, it would strengthen the argument to adapt baseline methods to the current setting and compare their performance with that of the proposed method, making the results more persuasive.

5. The final loss function includes two hyperparameters, making it crucial to conduct a sensitivity analysis to evaluate the method's robustness. However, the absence of such experiments raises concerns about the reliability and generalizability of the proposed approach.

6.  The assumption that all observed variables are confounders may be stringent and difficult to satisfy in practical scenarios.

7. The two-stage method presents several limitations, such as potential inefficiencies in estimation and sensitivity to the parameters selected at each stage, which may compromise the overall robustness of the results.

**Questions:**

See Weaknesses part.

---

> ### Author Response · Authors · 2024-11-20
>
> Thank you for your actionable review and your helpful suggestions! As you can see below, we have carefully revised our paper along with your suggestions. We display our extended theoretical results and new experiments in  **our updated paper** with major changes highlighted in **blue color**.
>
> ### Response to Weaknesses
> 1. **Limited theoretical contribution**:
> Thank you for allowing us to elaborate on the novelty of our theoretical contribution! Here, we would shortly like to summarize the three main theoretical contributions over existing works:
>
>     (i) **Applicability to arbitrary IVs and target bound width minimization**: We show that the existing bounds for discrete instruments from Lemma 1 can be applied to other instrument types (continuous, high-dimensional) by using **arbitrary** partitioning functions, enabling to transfer and generalize the bounds to new unconsidered settings such as Mendelian randomization (MR) or indirect experiments with complex nudges. While this may seem straightforward at first sight, to the best of our knowledge, we are **not aware of any prior work** considering that connection, i.e., even our **naive baseline leveraging $k$-means clustering has not been considered before**. Further, this finding allows us to develop the **new objective** of **directly targeting bound width minimization** during representation learning to learn optimal partitions (**Eq. (8)**). This leads to the two major theoretical contributions regarding optimized training:
>
>     (ii) **Stability by avoiding alternating learning**: A _straightforward_ implementation minimizing the bounds following Eq. (8) would require _alternating learning_. The reason is that, after **every update step** of $\phi(z)$, the quantities $\mu_\phi^a(x, \ell)$ and $\pi_\phi(x, \ell)$ are not valid for the updated $\phi$ anymore and would need to be **retrained** to ensure valid bounds. This is _computationally highly expensive_ and results in _unstable training and convergence problems_. However, our method circumvents these issues: by using our **novel Theorem 1**, we show that, while training $\phi(z)$, the quantities $\mu_\phi^a(x, \ell)$ and $\pi_\phi(x, \ell)$ can be **directly calculated**. \
>     For that, we can simply evaluate the nuisance functions, which only need to be trained **once** in the first stage. Therefore, **we avoid any need for alternating learning**, resulting in more **efficient and stable** training. Here, also note that Theorem 1 and its proof in Appendix A1 target **effective estimation** of our target quantities, and thus is orthogonal to the works about discrete instruments which aim for the **derivation** of bounds.
>
>    (iii) **Improved finite sample robustness**. Even using our stable training procedure from above, optimizing for Eq. (4) only yields valid bounds _in expectation on the population level_. However, if the discrete representation learning results in highly imbalanced marginal probabilities during training (i.e., $\mathbb{P}(\phi(Z)=\ell)$ is small for some $\ell$), this can result in _high estimation variance_ of the nuisance estimates and thus _unreliable bound estimates_. We show that more formally in our **Theorem 2** where we provide theoretical guarantees for the asymptotic behavior. In contrast, we avoid these problems: by using our theoretically motivated custom loss from Eq. 15 with the respective regularization from Eq. 14, we enforce _lower estimation variance_ during training and thus _more reliable bound estimates_.
>
>     **Action:** We further improved the explanation and presentation of the advantages of our method in our updated paper.
>
> 2. **Additional experiments for high-dimensional instruments**: Thank you for this remark! While we consider modeling three varying settings motivated by Mendelian randomization, we fully agree that our paper clearly benefits from additional experiments with even higher-dimensional instruments. **Action:** Thus, we added additional experiments for additional DGPs and with up to 100 dimensional instruments. The results are presented in our **new Appendix E (esp. E2)**. Here, _our method performs clearly more robustly while providing more informative bounds than the considered baselines_. For more information regarding performance evaluation and additional baselines, we kindly refer to our next two answers to points 3. and 4.:

---

> ### Author Response · Authors · 2024-11-20
>
> 3. **Effectiveness of our method**: Here, we would like to kindly emphasize that our method **does** show **substantial improvements over the baselines**. However, we fully agree that this improvement is not obvious at first sight and that we can improve our presentation. Here we would shortly like to summarize the challenges of benchmarking the performance of bound estimators in our setting and finally present our adjustments for improved presentation:\
> (i) Since for datasets 1 and 2 we model an underlying _continuous_ influence of the instrument on the treatment, we **cannot** approximate oracle bounds (but only the oracle CATE). Thus, while we can judge the validity of the bounds on the CATE level, we **cannot** directly judge the estimation of the bound width (i.e., it might be falsely overconfident). Instead, we can evaluate the robustness of our estimates (if they yield the same estimates under different settings). Here, our method shows clearly lower variation as represented by the MSD with **relative reductions of about $89$% and $33$%** compared to the baseline with about equal average bound width, indicating clearly more robust estimation of our method. **Action:** To further improve our presentation, we presented our metrics averaged over the runs in our **new Table 1** and report performance improvements explicitly.\
>  (ii) For dataset 3 and our new dataset 4, we can see **relative reductions in the MSD of about $70$% and $48$%**. Further, for datasets 3 and 4, we model latent discrete instrument groups, so that we **can** approximate oracle bound estimates and check for their coverage given our bound estimates. Recall that, for reliable decision-making, we would like to obtain **tight bounds** but only under the constraint that they yield **reliable coverage**. Thus, we should only compare the bound width of estimates with near-perfect coverage to the oracle bounds (e.g.., $\geq 95$%). Here, we again see that our method is superior by a large margin.\
>  **Action:** We introduced the new metrics to display the average bound width and MSE under reliable estimation in Sec. 6 under “Performance metrics”. We present the results in **our new Table 2 and in Appendix E**. Further, we show additional baselines (see also our following answer to point 4.) **Evidently, our method performs clearly best regarding robustness in estimation of tight valid bounds compared to the baselines.**
>
> 4. **Additional adapted baselines**: Thank you for this suggestion. While we argue that existing methods are not tailored to our setting, we agree that additional naively adapted baselines might further strengthen our motivation. Specifically, we adapted two additional baseline methods to our setting and compared their performance to our approach:\
> (i) **_DeepIV with bootstrapped confidence intervals_**: DeepIV is designed for high-dimensional instruments under point identification. We approximated 95% confidence intervals using bootstrapping, which accounts for statistical uncertainty but not for identifiability uncertainty due to violations of causal assumptions. This highlights why bound estimators like our method are necessary.\
> (ii) **_Bounds on discretized IVs_**: We discretized high-dimensional IVs using a simple grouping rule instead of learning partitions and estimated the bounds for discrete IVs. This allowed us to demonstrate the advantages of our tailored representation learning and the robustness of our method due to estimating fewer models.\
> **Action**: We reported the results in **Appendix E1 and E2**. We observe superior performance of our method compared to all baselines, empirically supporting the effectiveness of our method.
>
> 5. **Sensitivity over hyperparameters**: Thank you for this suggestion! **Action:** We included additional sensitivity analysis for hyperparameters $\lambda$ and $\gamma$ in **Appendix G**. Further, we also provide an interpretation and a guideline for selecting the parameters, as well as an extended discussion for the parameter $k$ in **Appendix F**. We shortly summarize our findings as follows: \
> As expected, **our method is robust** for reasonable values of the hyperparameters. $\gamma$ does not affect the bound size but can be optimized to reduce estimation variance, as mentioned in the motivation of our auxiliary guidance loss. $\lambda$ demonstrates the trade-off between tightness and variance and shows the importance of our regularization loss. Here, $\lambda$ can be increased to reduce the variance. In our experiments, the optimal tradeoff between reduced variance and bound tightness also results in optimal oracle coverage, showing the practicability of our regularization.

---

> ### Author Response · Authors · 2024-11-20
>
> 6. **Assumptions about observed variables**: Thank you for allowing us to clarify our assumptions about the observed variables $X$! Indeed the **only assumptions** we make for $X$ are the indepedencies in Assumptions 2 and 3. Importantly these assumptions only indicate that we **allow** for $X$ to be a confounder. However, note that it is also perfectly fine if $X$ has no effect on $A$, no effect on $Y$, or even both, and our bounds **still remain valid**. We only do not allow for $X$ to be correlated with $Z$ or to be a post-treatment variable. However, here we can show another advantage of using instruments instead of only adjusting for observed confounders: If we are unsure whether some subset of $X$ satisfies our assumptions, we can just **exclude** it from being used in our method since we explicitly **allow for unobserved confounding**. Depending on the setting, this might result in wider bounds, but they are still **guaranteed to be valid**. This is a **clear advantage** over normal confounder adjustment, which would result in misspecified estimates in such cases.
>
> 7. **Limitations of two-stage method**: We agree that our method has some limitations (e.g., it requires accurate estimation of the first-stage nuisance functions) and could be further improved such as by deriving doubly robust estimators. However, we would kindly like to emphasize that the two-stage procedure and the option to adjust the hyperparameters do not lead to inefficiencies but, instead, are the **main drivers to increase the robustness of the estimation** due to the following reasons.
>
>     (i) _Validity of the estimated bounds_: Given that our first-stage nuisance estimators are consistent, also the bounds will be consistent for the representation mapping $\phi$ during training. This is unlike approaches ignoring our Theorem 1, which would require alternating learning procedures. For every updated $\phi$, the representation-dependent nuisance estimates would need to be retrained which results in **highly unstable and inefficient learning**. Thus, leveraging the two-stage procedure enabled by Theorem 1, our method encourages **stable and computationally efficient** learning while yielding reliable bound estimates.
>
>     (ii) After training the first-stage nuisance estimators once, they can be reused for every new parameter constellation for learning the optimal partitions $\phi$ (e.g., vary $k$, $\lambda$, $\gamma$). This allows us to focus on properly tuning the first stage nuisance estimates once and then focus on the training $\phi$ in the second stage. This is a major advantage over methods such as the naive k-means baseline which requires (a) retraining of the clustering and (b) retraining of the representation-dependent nuisance functions for **every** different value $k$. Thus, for tuning and model selection, our method requires training **fewer models** which implies that our method is more robust and less sensitive to potential errors of estimated models.
>
>     (iii) By leveraging our objective to directly learn $\phi$ to minimize the bound width while regularizing for estimator variance, our method is **more robust against the choices of the parameters such as $k$**. In contrast, the naive baseline treats the partitioning and the bound estimation as separate problems, which results in less efficient and less robust estimation. This is also shown empirically by our experiments, especially by the clearly lower MSD of our method.
>
>     **Action:** We added a summary of the advantages of the two-stage method in our updated paper. In summary, _our two-stage method is designed to increase the robustness and effectivity of the estimated bounds in our considered setting._ However, we find that extending our method further (e.g., to provide provable guarantees from semi-parametric efficiency theory) would be an interesting research direction!

---

> > ### Comment · Reviewer_ePQ5 · 2024-11-25
> >
> > Thank you for your detailed response. However, I still have concerns regarding the clarity of theoretical innovation and the limitations in application. After thoroughly reviewing the discussion between you and all the reviewers, I have decided to maintain my score.

---

> > > ### Author Response · Authors · 2024-11-25
> > >
> > > Dear Reviewer ePQ5,
> > >
> > > Thank you for your response and for the time and thought you have dedicated to reviewing our work! We understand that you still have concerns regarding the clarity of our theoretical contributions and the limitations in application. In our updated paper and rebuttal, we have worked to address these points in different ways. We are eager to address any open points or suggestions you might have!
> > >
> > > Regarding the **theoretical contributions**, we (1) discussed the generalization to arbitrary IVs in Sections 4.1 and 4.2 and Appendix A1, (2) introduced a novel learning objective and a stable learning procedure in Section 4.2 and Appendix A.1, and (3) demonstrated improved finite sample robustness by controlling the tightness-variance trade-off in Sections 4.3 and 5, as well as Appendices A.2 and A.3.
> > >
> > > Regarding **applicability**, we (1) showed the improved robustness of the two-stage method and loss terms compared to baselines in Sections 5 and 6 and Appendix E, (2) highlighted applications, including Mendelian randomization and indirect experiments, in Section 1 and Appendix B, and (3) demonstrated that our approach relies on weaker assumptions compared to prior work in Sections 1 and 2 and Appendix B.
> > >
> > > Could you kindly let us know which aspects of these points or also other parts of the paper remain unclear or where you feel further clarifications or improvements would be most helpful? For instance, do you think additional explanations or examples would make our contributions clearer? We would be very grateful for any actionable suggestions or detailed feedback to help refine and improve our work.
> > >
> > > Thank you again for your thoughtful review and for the time and effort you’ve invested in this process. We sincerely appreciate your feedback and look forward to hearing from you.
> > >
> > > Best regards,
> > > The Authors

---

### Official Review · Reviewer_bx94 · 2024-10-31

**Soundness:** 3
**Presentation:** 3
**Contribution:** 2
**Rating:** 5
**Confidence:** 3

**Summary:**

Reliable estimation of treatment effects from observational data is crucial in many disciplines and is a challenge when unconfoundedness assumption in the causal inference literature is violated. In this paper, the authors leverage arbitrary instruments to estimate bounds on the conditional average treatment effect (CATE). First, the authors propose a novel approach for partial identification by mapping instruments to a discrete representation space such that valid bounds on the CATE are yielded. Second, the authors derive a two-step procedure that learns tight bounds using a tailored neural partitioning of the latent instrument space. Thus, instability issues due to numerical approximations or adversarial training are avoided. Third, the authors provide theoretical studies on the proposed approach. Finally, the authors perform extensive experiments to demonstrate the effectiveness across various settings.

**Strengths:**

This paper investigates the conditional average treatment effect (CATE), assuming the presence of unobserved confounders. Further, researchers encounter complex instrument information (e.g., continuous/high-dimensional), which makes the computation hard in many circumstances. The authors aim to develop estimators that can provide reliable analysis of the above assumptions. Unlike the existing literature, the authors establish bounding estimators of the target quantity and provide theoretical studies of the constructed estimators. A merit of the paper is that it is well-written, so readers can easily follow its notions.

**Weaknesses:**

There are some notational problems in the paper. For example:
1. Notational problems in Eqn. (10).
2. According to Eqn. (8) (or Eqn. (14)), $\phi$ should be a representation map, and $\Phi$ should be a set of all representations. It is quite confusing to write $\Phi(Z)=l$ (e.g., see line 323).
3. Please also state the explicit form of the constants $c$, $d$ in Theorem 2.
Please recheck the notations to avoid any confusions.

**Questions:**

Marks will be adjusted according to the replies. Below are some feedback/questions:
1. What is the point of building bounding estimators? As mentioned in the paper, existing literature focuses on point estimators. Indeed, we can build the confidence interval based on the point estimator. It is well-known that these are two concepts, but it is odds to build bounding estimators present in the paper.
2. Theorem 2 gives the asymptotic properties of the nuisance estimators $\hat{\mu}$ and $\hat{pi}$ when $\phi$ is arbitrary. It would be better to study the asymptotic properties of the nuisance estimators due to optimal $\phi^*$ after Eqn. (8).
3. It is strange to have $N_b$ in Algorithm 1 (see lines 387, 388).
4. The auxiliary guidance loss is not clear enough. It would be better if the authors could provide an explicit formula for the loss similar to bandwidth minimization loss and regularization loss.
5. The size of $k$ should affect the bounded estimators. Please demonstrate a scientific way to choose $k$. In the existing experiments, authors study results for different $k$. In any case, it seems that increasing $k$ would give better results. It should not be correct in general.
6. Could you provide the results of the following experiment:
assuming linear generative model, nonlinear ML methods are used to estimate the nuisance functions and then construct the corresponding bandwidth;
assuming a nonlinear generative model, linear ML methods are used to estimate the nuisance functions and then construct the corresponding bandwidth.
7. Would the constructed bounding estimators be doubly robust?

---

> ### Author Response · Authors · 2024-11-20
>
> Thank you a lot for your thorough review and helpful suggestions! We show **our extended theoretical results and new experiments** in  **our updated paper**. Therein, we highlighted major changes in **blue color**. We improved our paper as follows.
>
> ### Response to Weaknesses
> Thank you a lot for your careful checks of our notation! We worked over our paper again and made the following changes:
>
> 1. Thank you! **Action:** We addressed the notational problems in Eq. (10)
>
> 2. Thanks, you are correct: $\Phi$ should indeed present the set of all representations!
> **Action:** We adjusted our inconsistencies in notation to properly distinguish between the representation map $\phi$ and the set $\Phi$.
>
> 3. Thanks for this important remark! **Action:** To improve the readability of our **Theorem 2**., we included the explicit forms of the constants $c$ and $d$ as derived in our proofs in the appendix below the theorem in the main paper. Further, after incorporating the final edits and suggestions from the rebuttal, we will thoroughly recheck and fix any other appearing inconsistencies for the final version.
>
> ### Response to Questions
> 1. **Importance of bound estimators:** Thank you for this question. This is indeed an important point! The purpose of **bounding estimators** is to address **identifiability uncertainty** (=unreducible even with infinite $n$) — a fundamental limitation when point identification is impossible. This is the case when the additivity/homogeneity or relevance assumptions that are necessary for point identification with IVs do not hold. This is **distinct** from the role of **confidence intervals**, which quantify **statistical uncertainty** (=reducible to some extent) around point estimates if these assumptions hold. In extreme cases with large sample size $n$ but a violation of these assumptions, this can lead to **falsely tight (overconfident) confidence intervals** with close to no coverage because of misspecified point estimators and confidence intervals (i.e., **no guarantees**). However, we acknowledge, that while most work addresses the _derivation of such bounds_, only a few works have yet addressed the _effective estimation_ of such bounds, and are usually tailored for discrete IVs (e.g., [1], [2]). In our work, we extend this literature stream to incorporate also continuous and high-dimensional IVs. \
> **Action:** To empirically demonstrate the necessity of using bound estimators, in **our new Appendix E.1**, we implement an **additional baseline (DeepIV)** which is tailored for high-dimensional instruments when point identification can be ensured. This requires the **additional assumption** of additivity of the unobserved confounding, which usually cannot be ensured and is not needed for our method. For DeepIV, we can approximate confidence intervals using bootstrapping. Here, we approximate confidence intervals with a confidence level of $95$%, indicating an expected coverage of $95$% if assumptions were not violated. As expected, **DeepIV gives _falsely_ overconfident bounds** with low coverage of the true CATE and no coverage of the oracle bounds which emphasizes the **necessity for using proper bound estimators**.
>
> 2. **Importance of asymptotic nuisance properties for arbitrary $\phi$**: We agree that, at first sight, studying asymptotic properties for arbitrary $\phi$ might seem counterintuitive.  However, we would like to emphasize that the main motivation of Theorem 2 is **not** to study the asymptotic properties on the finally learned or even optimal bounds (e.g., to build confidence intervals on top of the bound estimates). Instead, we aim to yield **reliable final bound estimates** by already ensuring valid bound estimates for robustly updating $\phi$ **during training**. For that, we want to ensure that not only the final nuisance estimators and bounds have reduced variance, but all the nuisance functions should be estimated reliably for every update step to guarantee stable training. This explicitly includes encouraging reliable estimation of the nuisance functions also for **sub-optimal** $\phi$, e.g., during the early steps of the training phase.  As a beneficial consequence, since we encourage reliable estimation at all stages, the final bounds built on top of these nuisance functions after training will **also yield reliable estimates**.
>
> 3. Thank you for this remark! **Action:** We corrected Algorithm 1 to properly follow our notation (moved to **Appendix H** in our updated paper).
>
> 4. Thank you for this suggestion! **Action:** To improve the clarity of our method, we included an explicit formula for our auxiliary guidance loss in our paper in Eq. (17) as
> $$\mathcal{L}_{aux}(\theta) = - \frac{1}{n} \sum^n _{i=1} \sum^k _{j=1} \mathbf{1} [{\phi _\theta}(z_i) = j] log(p _\zeta(z_i)), $$
>
>     where $p_\zeta(z_i)$ is the predicted probability of assigning $z_i$ to discrete representation $j$ by the additional classification head.

---

> ### Author Response · Authors · 2024-11-20
>
> 5. **Role and selection guideline of parameter $k$**: Thank you for these interesting remarks! Indeed, $k$ plays an important role in the motivation of our method. Here, we first would like to explain that a major benefit of our method is that it directly targets robust estimation over varying values of $k$, thus mitigating the importance of selecting an optimal $k$ which is unlike the unstable naive baseline.  Empirically, in our experiments, this is demonstrated by **lower variance and stable behavior over varying** $k$, especially visible in the **low values of MSD**. This is due to the combination of learning flexible representations tailored to minimize bound width (allowing to estimate tight bounds already for low $k$) while ensuring reliable estimates of the nuisance functions in the second stage by using our regularization loss in Eq.(16) (ensuring robust behavior also for higher $k$). Note that this robustness is **especially beneficial when applying our method to real-world settings** in causal inference. In real-world settings, hyperparameter tuning and model evaluation are **not straightforwardly possible** because oracle CATE or oracle bounds are not known. Thus, such **robustness** against suboptimal selection of hyperparameters such as $k$  **is crucial**, especially in high-stakes decision-making such as medicine.
>
>     _How $k$ affects the estimators_:  On a population level, the bounds get tighter with growing $k$. This follows from Theorem 1, since using more $k$ increases the flexibility of $\phi$. However, from a finite sample perspective, the variance of the nuisance estimators (and thus, also likely of the final bounds), will increase for increasing $k$. To avoid this, we designed our regularization loss in Eq. (16). However, as an interesting note, for some fixed (not too large) $\lambda$, the penalization term in Eq. (16) will also increase with increasing $k$, which yields an automated stabilization for higher $k$. This is also shown in our experiments where higher values of $k$ do not necessarily result in a higher variance. However, this behavior can vary for different datasets or over different values of $\lambda$. **Action:** We also included a sensitivity over $\lambda$ for given $k$ showing the tradeoff between bound tightness and variance in **Appendix G**.
>
>     _Guideline to choose $k$_: As stated above, our method mitigates the relevance of choosing an optimal $k$. Nevertheless, we also included practical recommendations on how to choose $k$ in real-world settings without access to oracle bounds in **Appendix F**. We can summarize them as follows: **Approach 1: Expert informed**. In some medical applications, physicians might already know or make an educated guess about a number of underlying clusters of patient characteristics such as genetic variants. For instance., this is a common assumption in subgroup identification subgroup identification or latent class analysis in medicine where patient groups are characterized by having similar responses to treatments or showing similar associations with diseases. Thus, no data-driven approach is necessary here. **Approach 2: data-driven for hypothesis confirmation**. Often, physicians are interested in whether some treatment or exposure has a positive or negative effect (=lower bound > 0 or upper bound < 0) for at least some observations $x$. Thus, $k$ can be selected by increasing $k$ until such an effect can be observed while holding the variance minimal. Then, the variance can be approximated (e.g., by bootstrapping to test for the reliability of the corresponding bound model and its effect). This offers a practical way to choose the parameter.
>
>     **Action:** We included a discussion about the role of $k$ and practical recommendations on how to choose $k$ in real-world settings in **Appendix F**.
>
> 6. **Additional experiments:** Thank you. We are happy to provide more experiments for different DGPs and different nuisance models. Again, the new experiments demonstrate the effectiveness of our proposed method. **Action:** We included experiments with a linear data-generating process and non-linear ML estimators, as well as a non-linear DGP with linear models for nuisance estimation in **Appendix E3**. We observe, as expected for a linear DGP, that**our method still performs robustly** and is highly effective. In the other setting with linear models but non-linear DGP, we cannot guarantee the validity of the bounds without flexible estimation (as expected). This shows the importance of leveraging NNs as a non-linear ML method in our approach. Nevertheless, **we still outperform the naive baseline**, especially with respect to the coverage, which demonstrates empirically the **robustness of our method even when using under-complex models**.

---

> ### Author Response · Authors · 2024-11-20
>
> 7. **Doubly robustness**: Thanks for this interesting question! In principle, our method is not doubly robust with respect to misspecification of our **first step nuisance estimators**, i.e., $\hat{\mu}$, $\hat{\pi}$, $\hat{\eta}$ need to be consistent (that is why we use NNs). However, under this assumption, it follows from Theorem 1 that the bounds for arbitrary $\phi$ are consistent. This is important (i) for reliable and stable learning and (ii) for yielding robustness of varying $k$. This is a major advantage over using alternating learning, where the nuisance estimates for every updated $\phi$ need to be re-estimated, which can easily result in misspecification during training. This is also a clear benefit over the naive method with $k$ means clustering which requires retraining of the nuisance estimates for every new selection of $k$, meaning that the naive baseline can essentially become unstable when evaluating multiple and high-values of $k$. One can also see this in our experiments by the high MSD. \
> **Summarized:** While our method does not give theoretical properties of doubly robustness, it nevertheless requires **estimating fewer models** and provides **more robust estimates**. However, we find that extending our method to provide doubly or multiply robust properties would be an interesting direction for future research!
>
> ### References
> [1] Schweisthal, J., Frauen, D., van der Schaar, M., & Feuerriegel, S. (2024). Meta-Learners for Partially-Identified Treatment Effects Across Multiple Environments. In ICML.
>
> [2] Levis, A. W., Bonvini, M., Zeng, Z., Keele, L., & Kennedy, E. H. (2023). Covariate-assisted bounds on causal effects with instrumental variables. arXiv preprint arXiv:2301.12106.

---

> > ### Comment · Reviewer_bx94 · 2024-11-26
> >
> > Thanks for your replies. Most questions are addressed. Indeed, I am struggling if the theoretical studies are enough, especially points 2 and 7. Conservatively, I will keep my score.

---

> ### Author Response · Authors · 2024-11-27
>
> (1/2)
>
> Dear Reviewer bx94,
>
> Thank you for your response and the effort you have dedicated to the review process, we are happy that we could address most of your questions! Further, we understand that you are still uncertain regarding our theoretical studies especially in our answers to point 2 and 7 of your review. We are grateful that you give us the opportunity to further address these points and we are confident that we can resolve any remaining concerns. Below, we would like to address both of your points, and uploaded a revised version of our paper with additional changes in **red** to highlight these from the changes in blue done during our first revision.
>
> **(Q2)** In our answer to your **Question 2 (Asymptotic properties of the nuisance estimators in Theorem 2**), we emphasized that we are indeed interested in the properties for _arbitrary_ representations $\phi$ (and not only for optimal $\phi^*$) to motivate our regularization loss from Eq. (16), resulting in more stable updates and more relaile bound estimates. However, after thoroughly checking our paper again, we think we might have found a potential reason for confusion. In our original version, we stated the tightness-bias-variance trade-off of our Lemma 1 for bound estimators $\hat{b}^+_ {\phi*} (x)$ for _optimal_ $\phi^*$. However, importantly, this also holds for estimators $\hat{b}_{\phi}^+(x)$ of _arbitrary_ $\phi$, which is clearly more consistent with our narrative of the previous and following parts of our paper. We apologize for this confusion and corrected this in the current version.
>
> Further, in our previous answer, we missed to mention another point (apologies!): Since the properties shown in our Theorem 2 hold for arbitrary $\phi$, this implies that these properties also directly hold for the finally learned or even optimal $\phi^\ast$ as requested in your original question. Importantly, this means that not only the nuisance estimates _during_ training, but also the once used finally _after_ training to calculate the bounds, both will have reduced variance. **Thus, we not only ensure more stable training, but also more reliable final estimates**. _We added this important detail to our paper_.
>
> **Action:** In our updated paper, we clarified that our Lemma 1 and the corresponding proof hold for arbitrary $\phi$. Further, we added a footnote in Sec. 4.3. to emphasize that Lemma 1 and and Theorem 2 hold as well for arbitrary and optimal representations and state its beneficial implications.

---

> ### Author Response · Authors · 2024-11-27
>
> (2/2)
>
> **(Q7)** Thank you for the opportunity to extend on our answer about your question about **doubly robustness**. We fully agree that doubly/multiply robust properties is a highly interesting research direction. However, deriving such an estimator in our setting of partial identification with continuous or high-dimensional IVs is **way more complex** than for example for settings with point-identification or discrete instruments, **and might be even impossible with the current statistical toolkits**. To explain this in more detail, and since we think this is an really interesting direction, we now provide an additional discussion around doubly robustness and its challenges in our considered setting in our **new Appendix I**.  We shortly summarize our discussion as follows:
>
>
> To derive doubly or multiply robust estimators in the traditional way, we would need to derive the efficient influence function of the causal quantity we want to estimate (see, e.g., [1] for the DR-learner of the CATE). In our setting, these quantities are the partial identifaction bounds for the CATE. However, under our assumptions 1-3, _no closed-form solution exists_ for the bounds for the CATE for general IVs (i.e., continuous or high-dimensional). Instead, we can only formalize the identification of the bounds as a constraint optimization problem (see e.g., [2]), as we do in Eq (1). Since the constrained optimization problem is not pathwise-differentiable, the current statistical efficiency theory used for deriving doubly robustness is not applicable. Thus, deriving doubly robust estimators without a closed-form solution is not solvable with the usual toolkit and highly non-trivial. Further, we are _not aware of any other approaches_ to solve such or a similar setting. However, if you are aware of any related work we might have missed, we would be  very interested and happy to have a closer look!
>
> To this end, one advantage of our method is that we implicitly learn tailored discretizations of the IVs. In principle, _after learning the discretization_, we could apply the recently developed meta-learners for bounds with _discrete_ IVs of [3], including their doubly robust bounds learner, to estimate the final bounds. However, this _does not guarantee doubly robustness for the original bounding problem_ but only on the already learned representations. Further, since we would not use the optimized nuisance estimates from our approach for bound calculation but, instead, require estimation of additional models, this might easily result in increased variance and leads to computational overhead. Therefore, we would not recommend this approach.
>
> **Action:** We added a more extensive discussion for both points to our revised PDF; kindly see our **new Appendix I**.
>
> If you should have any remaining concerns regarding our theoretical studies or any other suggestions on how to improve the theory or other parts of the paper, we would be very grateful to address them in the remaining discussion period and incorporate them into the final version of our paper!
>
> We truly appreciate your feedback and look forward to your response.
>
> Best regards,
>
> The Authors
>
> **References**
>
> [1] Kennedy, E. H. (2023). Towards optimal doubly robust estimation of heterogeneous causal effects. Electronic Journal of Statistics, 17(2), 3008-3049. \
> [2] Kilbertus, N., Kusner, M. J., & Silva, R. (2020). A class of algorithms for general instrumental variable models. NeurIPS. \
> [3] Schweisthal, J., Frauen, D., van der Schaar, M., & Feuerriegel, S. (2024). Meta-Learners for Partially-Identified Treatment Effects Across Multiple Environments. In ICML.

---

### Official Review · Reviewer_EWM3 · 2024-11-01

**Soundness:** 3
**Presentation:** 3
**Contribution:** 2
**Rating:** 5
**Confidence:** 3

**Summary:**

This paper proposes a partial idetification approach of the conditional average treatment effect (CATE) that allows for arbitrary instruments. In particular, the proposed appraoch first maps the intstruments to a discrete representation space, and then leverages partial identification results for discrete intruments to bound the CATE. The representation mapping is learned by minimizing the length of the bounding interval while stabilizing finite sample performance. The authors also provided a neural implementation of the proposed method and performed simulation experiments.

**Strengths:**

- I appreciate the clear comparison with existing works.
- The paper provided clear description of a neural implementation of the proposed approach, and there is some innovation in the custom loss function.

**Weaknesses:**

Overall, the theoretical contribution of the paper appears limited (some of them are very straight forward applications of the existing results for discrete instruments) and may benefit from further exploration.
- The paper only provided finite samle guarantees for the second stage nuisance functions, but not for the final bounds, which might be more useful (for e.g. guaging the quality of the estimation) and easier to interpret. For instance it would be nice to have some theoretical results on the coverage (or scaled coverage - coverage/length of interval) of the estimated bounds.
- The number of partitions ($k$) is an important hyperparameter, but the paper did not provide anytheoretical results on how the estimation error or the tightness of the bounds would scale with $k$.

**Questions:**

- Even the experiements in the high dimensional settings had only 20 dimensional instruments. I am not familar with Mendelian randomization with SNPs, so I wonder if this falls into the normal range of such applications.
- Is the bound from Theorem 1 sharp?

---

> ### Author Response · Authors · 2024-11-20
>
> Thank you for your positive and actionable review! We took all your comments at heart and improved our paper as follows. We display our extended theoretical results and new experiments in **our updated paper** with major changes highlighted in **blue color**.
>
> ### Response to Weaknesses
>
> **Limited theoretical contribution**:
> Thank you for allowing us to elaborate on the novelty of our theoretical contribution! Here, we would shortly like to summarize the main three contributions over the existing results for discrete instruments:
>
> **Applicability to arbitrary IVs and target bound width minimization**: We find that the existing bounds for discrete instruments from Lemma 1 can be applied to other instrument types (continuous, high-dimensional) by using **arbitrary** partitioning functions, enabling to transfer and generalize the bounds to new unconsidered settings such as Mendelian randomization (MR) or indirect experiments with complex nudges. While this may seem straightforward at first sight, it is highly non-trivial. To the best of our knowledge, we are **not aware of any prior work** considering that connection, i.e., even our naive baseline leveraging k-means clustering has not been considered before and is thus novel. Further, this finding allows us to develop the **new objective** of **directly targeting bound width minimization** during representation learning to learn optimal partitions (**Eq. (8)**). Accordingly, we provide two major theoretical contributions regarding optimized training:
>
> a. **Stability by avoiding alternating learning**: A naive implementation minimizing the bounds following Eq. (8) would require _alternating learning_. The reason is that, after **every update step** of $\phi(z)$, the quantities $\mu_\phi^a(x, l)$ and $\pi_\phi^a(x, l)$ are not valid for the updated $\phi$ anymore and would need to be **retrained** to ensure valid bounds. This is _computationally highly expensive_ and results in _unstable training and convergence problems_. However, our method circumvents these issues: by using our **novel Theorem 1**, we show that, while training $\phi(z)$, the quantities $\mu_\phi^a(x, \ell)$ and $\pi_\phi(x, \ell)$ can be **directly calculated**.
> For that, we can simply evaluate the nuisance functions, which only need to be trained **once** in the first stage. Therefore, **we avoid any need for alternating learning**, resulting in more **efficient and stable** training. Here, also note that Theorem 1 and its proof in Appendix A1 target **effective estimation** of our target quantities, and thus is orthogonal to the works about discrete instruments which aim for the **derivation** of bounds.
>
> b. **Improved finite sample robustness**. Even using our stable training procedure from above, optimizing for Eq. (4) only yields valid bounds _in expectation on the population level_. However, if the discrete representation learning results in highly imbalance marginal probabilities during training (i.e., $\mathbb{P}(\phi(Z)=\ell)$ is small for some $\ell$), this can result in _high estimation variance_ of the nuisance estimates and thus _unreliable bound estimates_. We show that more formally in our **Theorem 2**. In contrast, we avoid these problems: by using our custom loss from Eq. 15 with the respective regularization from Eq. 14, we enforce _lower estimation variance_ during training and thus _more reliable bound estimates_. We elaborate on this in more detail in the following, addressing both of your valid and interesting points in the following: (**Action:** We improved the presentation in our paper to explicitly spell out our contributions.)

---

> ### Author Response · Authors · 2024-11-20
>
> - **Finite sample guarantees**: We agree that we only provide finite sample guarantees for the nuisance functions instead of the final bounds. However, we would like to emphasize that the main motivation of Theorem 2 is **not** to construct confidence intervals or results on the width of the finally learned bounds. Instead, we aim to yield **reliable final bound estimates** by already ensuring valid bound estimates for robustly updating $\phi$ **during training**. For that, we want to ensure that not only the “sparse signal” of the bounds / bound-width (i.e., a minimum of combinations of the nuisance functions) has reduced variance, but all the nuisance functions should be estimated reliably for every update step to guarantee stable training. As a consequence, the final bounds built on top of these nuisance functions after training will **also yield reliable estimates**.
>
>   Further, we fully agree that final sample guarantees for the final bounds or bound width (e.g., to build confidence intervals) would be an interesting direction. However, to the best of our knowledge, we are not even aware of any works providing such results for bounds on the CATE in the simpler setting of discrete IVs and we leave this to future research. \
>   **Action:** We mention the previous point as a suggestion for future research in our extended “Limitations” section, especially for settings that are simpler than ours such as discrete IVs.
>
> - **Role of the number of partitions ($k$)**: One major advantage of our method is that our method is clearly less sensitive to the parameter $k$ than, for example, the naive baseline. Empirically, we demonstrate this in our experiments by lower variance and stable behavior over varying $k$, especially visible in the low values of MSD. This is due to the combination of learning flexible representations tailored to minimize bound width (allowing us to estimate tight bounds already for low $k$) while ensuring reliable estimates of the nuisance functions in the second stage by using our regularization loss in Eq.(16) (ensuring robust behavior also for higher $k$). Note that the robustness of our method is especially beneficial when applying our method to real-world settings in causal inference. In real-world settings, hyperparameter tuning and model evaluation are not directly possible because oracle CATE or oracle bounds are not known. Thus, such robustness against suboptimal selection of hyperparameters such as $k$  is crucial, especially in high-stakes decision-making such as medicine. However, although our method reduces the importance of the optimal selection of $k$, we agree that further theoretical analysis could be interesting.
>
>   As stated above, the exact theoretical derivation of the behavior of our method is highly non-trivial also for varying values of $k$, and also strongly depends on the true (unknown) data-generating process. Nevertheless, we are happy to provide the following additional insights (see our **new Appendix F**):
>
>   **Estimation error for different $k$**: The hyperparameter $\lambda$ controls the regularization loss in Eq. (16), i.e., it tries to maximize $\hat{p} =\hat{\mathbb{P}}(\phi_{\theta}(Z)=\ell) > \varepsilon$ for all $\ell \in 1, …, k$. Thus, if we choose $\lambda$ high enough, then we enforce that $\hat{p} = 1/k$ for all $\ell \in 1, \ldots, k$. Plugged into Theorem 12, the asymptotic variances for the nuisance estimators are $k \left(\frac{{Var}(g(Z) \mid {\phi}(Z) = \ell)}{c} + d \right)$ for  $\hat{\mu}^a_\phi (x, \ell)$, and $k \left({Var}(h(Z) \mid {\phi}(Z) = \ell)\right)$ for $ \hat{\pi}_{\phi}(x, \ell)$, respectively. Thus, for large enough $\lambda$, the variance of the nuisance estimators (and, thus, also likely of the final bounds) will increase for increasing $k$. However, as an interesting side note, for a fixed (not too large) $\lambda$, the penalization term in Eq. (16) will also grow with growing $k$ due to the same reason, which yields an automated stabilization for higher $k$. This is also shown in our experiments where higher values of $k$ do not necessarily result in a higher variance.

---

> ### Author Response · Authors · 2024-11-20
>
> -  [continued] **Role of the number of partitions ($k$)**
>
>     **Bound tightness for different $k$**: On a population level, the bounds get tighter with growing $k$. This follows straightforwardly, from Theorem 1, since using more $k$ increases the flexibility of $\phi$. While the exact bound width is highly non-trivial, we can use results from [1] about bounds for the CATE with discrete instruments to give some intuition. Specifically, in our setting, for some $x$, the CATE is bounded by
>
>    $ b^+ _\phi (x) - b^- _\phi (x) $
>
>    $\leq \min_{l, m} [ (s_2 - s_1) (2 - \pi_\phi (x, \ell) - (1- \pi_\phi (x, m))) ] $
>
>    with $\ell, m \in \{1, …, k\}$. This has two major implications. First, If for some $x$, $\phi$ is learned such that $\phi(x, \ell)$ is close to $1$ for some $l$ and $\pi_\phi(x, m)$ is close to $0$ for some $m$, the bound width is close to zero (“point identification”). Second, if the optimal partitioning function $\phi$ is the same for all $x$ (implying $b(x) = b$), then setting $k=3$ can be sufficient to yield the tightest bounds. This is because by using a flexible network for $\phi$, the partitions can be learned such that partition 1 yields propensity scores as close as possible to zero (as the data allows), partition 2 yields propensity scores as close as possible to 1, and partition 3 contains all $z$ resulting in propensity scores between those values. Note, however, that this is only valid in population but can result in highly unreliable estimation in finite sample data.
>
>    While we again agree that investigating these behaviors in more detail would be really interesting, we are also not aware of any works examining these properties for discrete instruments and we leave this to future work.
>
>    **Action:** We added a new discussion where we elaborate on the practical considerations regarding the role of $k$ (see our **new Appendix F**). Therein, we emphasize that our method is robust for different choices of $k$, which is a crucial advantage in causal ML where hyperparameter tuning is notoriously difficult and which should thus be seen as a strength of our method.
>
> ### Response to Questions:
> - **Dimensionality of IVs in MR**: Thanks for this question! Indeed, in Mendelian randomization, the number of IVs / SNPs can vary a lot depending on the setting. In many studies, the SNPs are already aggregated into an onedimensional allele score [2], which we analyze in datasets 1 and 2. When using the raw SNPs, the range can vary a lot. [3] provide a great summary of various applications with different numbers of SNPs. Here, it is common that the dimensions range between 1 and 100, but there also exist some studies with up to 696 considered SNPs. \
> **Action:** To show the validity of our method in more high-dimensional settings, we **added additional experiments with 100-dimensional IVs**. The results are in our **new Appendix E2**. We observe that our method has a robust performance, even in higher-dimensional settings.
>
> - **Sharpness of bounds**: This is an interesting question! In our method, we make use of the bounds of the CATE for discrete IVs as, e.g., used in [1]. Since these discrete bounds are not necessarily sharp, our bounds can also not be guaranteed to be sharp (but they are always valid). This also holds for $k \to \infty$ (even though the bounds become tighter for larger $k$ in population as explained above). However, under certain additional assumptions, it can be shown that these bounds for discrete IVs are tight for the average treatment effect (ATE). For a nice summary, we refer to [4]. We thus leave the development of sharp bounds of the CATE for discrete and complex IVs to future research. **Action:** We list the previous challenge as a question for future research in our extended “Limitations” section.
>
> ### References:
>
> [1] Schweisthal, J., Frauen, D., van der Schaar, M., & Feuerriegel, S. (2024). Meta-Learners for Partially-Identified Treatment Effects Across Multiple Environments. In ICML.
>
> [2] Burgess, S., & Thompson, S. G. (2013). Use of allele scores as instrumental variables for Mendelian randomization. International journal of epidemiology, 42(4), 1134-1144.
>
> [3] Pierce, B. L., Kraft, P., & Zhang, C. (2018). Mendelian randomization studies of cancer risk: a literature review. Current epidemiology reports, 5, 184-196.
>
> [4] Swanson, S. A., Hernán, M. A., Miller, M., Robins, J. M., & Richardson, T. S. (2018). Partial identification of the average treatment effect using instrumental variables: review of methods for binary instruments, treatments, and outcomes. Journal of the American Statistical Association, 113(522), 933-947.

---

> ### Comment · Reviewer_EWM3 · 2024-11-23
>
> Thank you for the detailed discussions and running the extra experiments! However, I am still concerned with the theoretical contributions of this work since this is mainly a proof of concept paper, therefore I will keep my score.

---

> ### Author Response · Authors · 2024-11-25
>
> Dear reviewer EWM3,
>
> Thank you for your response and for the time and effort you have dedicated to reviewing our paper. We sincerely appreciate your feedback!
>
> Here, we would kindly like to clarify that **our theoretical contributions** focus on the effective estimation of the bounds as stated in our objective in Eq. (14) by (i) avoiding alternating learning via leveraging our Theorem 1, and (ii) further stabilizing learning via our custom loss function in Eq. (18) motivated by our Theorem 2. However, importantly, we do _not_ aim to provide theoretical results for finite sample guarantees for the estimation error of the final bounds, bound width, or coverage, which is a different and highly challenging direction. _To the best of our knowledge, this has not been achieved or even explicitly targeted in prior works on partial identification of the CATE, even not for more simplified settings (e.g., binary or discrete instruments). While we agree that pursuing this is an important research direction, it is outside the focus of our current paper._
>
> Further, we agree that our work could be considered a proof of concept. However, we do not believe this needs to be a reason for rejection. On the contrary, this is why we targeted ICLR as a conference with a methodological focus. As such, our contribution over existing works is not only deriving bounds in a more complex setting but also proposing a procedure for robust estimation. Additionally, the setting we address—estimating partial identification bounds for the CATE with complex instruments—has not been directly targeted before. Most existing methods for this setting rely on _untestable and often unrealistic assumptions_. By offering an alternative that avoids these strong assumptions, we believe our approach provides an important foundation for future research and could provide a starting point for discussions in this underexplored area.
>
> We kindly seek your feedback on how to improve our work and remedy any remaining concerns. If you are aware of _related works we might have overlooked_ (e.g., for finite sample errors for bound estimates of the CATE), or if you have _additional suggestions on how to further improve the theoretical contribution_ of our paper in another way, we would be very grateful for your guidance. We are eager to explore them to further strengthen our paper!
>
> Thank you again for your thoughtful review and the time you’ve invested in this process! We truly appreciate your feedback and remain committed to further improving our work.
>
> Best regards,
>
> The Authors

---

### Official Review · Reviewer_hFWX · 2024-11-05

**Soundness:** 3
**Presentation:** 4
**Contribution:** 3
**Rating:** 8
**Confidence:** 3

**Summary:**

This paper provides a method for the partial identification of treatment effects when potentially high-dimensional instruments belonging to a broad class of domains are available. To overcome the potential of overly conservative bounds, it proposes a discrete embedding scheme for the instruments. It provides theoretical results for the validity of the proposed bounds and for estimator efficiency. It provides an empirical evaluation of the method using a well motivated simulated data generating process.

**Strengths:**

This paper, to my knowledge, provides a novel solution to a well motivated problem. It provides thorough theoretical results and adequate empirical evaluation. The writing quality and organization is excellent. The availability of high dimensional instruments is common in many settings beyond the medical domain, so I believe the potential for impact is high.

**Weaknesses:**

A discussion on potential applications outside of medicine may help to increase the visibility of this work. Perhaps by extending Appendix B.

**Questions:**

I have no further questions.

---

> ### Author Response · Authors · 2024-11-20
>
> Thank you a lot for your positive feedback and constructive input! We are really glad to hear that you liked our paper! It’s always rewarding to know that our work has been well-received.  We further improved our paper to incorporate your feedback as explained in the following.
>
> ### Response to Weaknesses:
> Thank you for your valuable input! We fully agree that the considered setting of continuous or high-dimensional instruments and our proposed method are relevant for applications beyond medicine.
>
> **Action:** We extended **Appendix B** to further explain and show in which settings our method could generate added value. For that, we extended the section on indirect experiments to show in more detail how our method can be used for evaluating causal effects in domains such as public health, marketing, and social sciences, and provided concrete examples.
>
> If you have any additional remarks or if any questions should arise during the discussion period, we would be very happy to address them!

---

> > ### Comment · Reviewer_hFWX · 2024-11-26
> >
> > Thank you for your response. I have looked at the changes in the updated paper, and read the reviews of and responses to the other reviewers. I am comfortable maintaining my recommendation for acceptance.

---

### Author Response · Authors · 2024-11-20

Thank you very much for the constructive evaluation of our paper and your helpful comments! We addressed all of them in the comments below. Furthermore, we uploaded an updated version of our paper and marked all significant changes to the original version in **blue color**.

Our **main improvements and clarifications** are the following:
- **Novel contributions**: We clarified our contribution as compared to previous work and now explicitly spell out how our work is novel.\
(i) We show that we can transfer the bounds for discrete instruments to our more complex setting such that they are valid for arbitrary mappings $\phi$ and clarify that also our optimization objective of learning representations to learn tight bounds has not been considered before.\
(ii) We elaborate on how _our Theorem 1 is novel and non-trivial_. Our Theom 1 allows for more effective, stable, and robust training by avoiding a straightforward implementation of our objective that would require alternating learning.\
(iii) We further explain how our finite sample perspective from Theorem 2 does not only focus on guarantees of the final bounds. Instead, we demonstrate that it is used to motivate our regularization loss to further robustify the bound evaluation already during training of $\phi$, which then also results in robust estimation of the final bounds.

- **Better writing & more details**: We **improved the presentation and motivation** of different parts in our paper:\
(i) We improve inconsistencies in our notation and explicitly add expressions such as constants in the proofs and the form of the auxiliary loss\
(ii) We discuss the role of the parameter $k$ and show that our method is evidently more robust against suboptimal selection against the baseline. Further, we added guidelines on how to select $k$.\
(iii) We improve the presentation of our experimental results. For that, we display new metrics focusing on the tradeoff between tightness and reliable estimation to further point out the robust estimation and clear performance gains of our method.

- **New experiments**: We **added new experiments to show the robustness of our method in various challenging settings**. Specifically, we provide new experiments in our **new Appendix E** for:\
(i) a new data set with 100-dimensional instruments (see **Appendix E.2**) including new baselines (see **Appendix E.1**),\
(ii)  ablations with different DGPs and models for nuisance estimation (see **Appendix E.3**), and\
(iii) sensitivity analysis over $\lambda$ and $\gamma$ (see our **Appendix G**). In all experiments, _our method performs robustly over varying scenarios_, showing the added value of our approach for settings when instruments are continuous or potentially high-dimensional.

We highlight all changes in the **updated PDF** in **blue color**. We specifically incorporated all changes (marked with **Action**) into the updated version of our paper. Given these improvements, we are confident that our paper provides valuable contributions to the causal machine learning literature and is a good fit for ICLR 2025.

---

### Author Response · Authors · 2024-12-02

Dear Reviewers,

Thank you again for your thoughtful feedback and engagement. We are confident that we have addressed all the points raised in your reviews and incorporated the suggested improvements into the revised manuscript in blue and red color.

We noticed that some reviewers have not yet responded to our latest updates and responses. Please let us know if there are any remaining questions or concerns — we would be more than happy to clarify further. Otherwise, if you feel our responses have addressed your concerns satisfactorily, we would kindly appreciate it if you would consider updating the rating.

We appreciate your time and effort and look forward to any additional feedback you might have.

Best regards,

The Authors

---

### Meta-Review · Area_Chair_Jh4r · 2024-12-21

**Metareview:**

This paper tackles the problem of bounding treatment effects (i.e. partial identification) when there is unobserved confounding in observational data. Here, the paper leverages instruments to estimate bounds on the conditional average treatment effect (CATE). This is done by first transforming instruments into a discrete representation space via the Gumbel Softmax trick followed by a two-step procedure to learn bounds on the CATE by partitioning the representation space. The work proposes using a loss function to learn the mapping by simultaneously minimizing the length of the bounding interval while stabilizing finite sample performance. The methodology was tested on simulated data.

Both the problem setting and the idea are interesting; but at the end of the discussion period this came down to a borderline decision. I closely followed the discussions and rebuttal. I think the authors did an excellent job of responding to the reviewers' questions but my reading of the discussion and responses was that the reviewers felt the manuscript in its current form read more like a proof of concept rather than a strong standalone piece of work either theoretically or empirically. Such a paper is reasonable but the reviewers did not find common ground on whether the idea alone, with simple experiments, sufficed for publication.

The authors clarify that their their major goal is "*not* to derive new identification bounds or new theoretical results for finite sample guarantees. Instead, the main contribution is to "provide a novel stable learning procedure in an important widely overlooked setting and to show its strong performance". Section 6 in the paper starts off by saying that no fair comparison is possible because there is not other work tailored to this setting. To that end, one suggestion to strengthen the manuscript is with further incorporation of baselines and empirical results in similar, even if not identical settings. For example, Appendix D1 in https://arxiv.org/abs/2202.10806 contains a semi-synthetic experiment based on Mendelian randomization that could form a point of comparison to the work. Similarly, the line of work on partial identification with neural causal models (https://arxiv.org/abs/2405.15673v2) also contains experiments on binary data (e.g. Table 1 in Appendix E https://arxiv.org/abs/2210.08139); this too could form yet another point of comparison. These would strengthen the claims around the efficiency of the estimator. Theoretically, an understanding of how the hyperparameter k influences the bounds would also improve the work. I encourage the authors to resubmit with improvements along the aforementioned axes.

**Additional Comments On Reviewer Discussion:**

hFWX felt positively about the work and maintained their score following the rebuttal. The authors made several changes to the manuscript adding several appendices from the initial submission; I believe the additions significantly improved the paper but bx94 and EWM3 continued to have concerns about whether the theoretical contributions sufficed for this work to stand on its own. ePQ5 does not respond to the rebuttal by the author (though I felt their questions were adequately addressed). I encourage the authors to think about how to take the comments herein, revamp the manuscript with a focus either on the theoretical aspects of the work with better clarity on why it is important and stands on its own or to expand on the empirical results (with specific suggestions on how to do so above).

---

### Decision · Program_Chairs · 2025-01-22

Reject